# Crowdsourcing via Pairwise Co-occurrences: Identifiability and Algorithms

**Shahana Ibrahim**
School of Elect. Eng. & Computer Sci.
Oregon State University
Corvallis, OR 97331
ibrahish@oregonstate.edu

**Xiao Fu**∗
School of Elect. Eng. & Computer Sci.
Oregon State University
Corvallis, OR 97331
xiao.fu@oregonstate.edu

**Nikos Kargas**
Department of Elect. & Computer Eng.
University of Minnesota
Minneapolis, MN 55455
kaga005@umn.edu

**Kejun Huang**
Department of Computing & Info. Sci. & Eng.
University of Florida
Gainesville, FL 32611
kejun.huang@ufl.edu

## Abstract

The data deluge comes with high demands for data labeling. Crowdsourcing (or, more generally, ensemble learning) techniques aim to produce accurate labels via integrating noisy, non-expert labeling from annotators. The classic Dawid-Skene estimator and its accompanying expectation maximization (EM) algorithm have been widely used, but the theoretical properties are not fully understood. Tensor methods were proposed to guarantee identification of the Dawid-Skene model, but the sample complexity is a hurdle for applying such approaches—since the tensor methods hinge on the availability of third-order statistics that are hard to reliably estimate given limited data. In this paper, we propose a framework using pairwise co-occurrences of the annotator responses, which naturally admits lower sample complexity. We show that the approach can identify the Dawid-Skene model under realistic conditions. We propose an algebraic algorithm reminiscent of convex geometry-based structured matrix factorization to solve the model identification problem efficiently, and an identifiability-enhanced algorithm for handling more challenging and critical scenarios. Experiments show that the proposed algorithms outperform the state-of-art algorithms under a variety of scenarios.

## 1 Introduction

**Background.** The drastically increasing availability of data has successfully enabled many timely applications in machine learning and artificial intelligence. At the same time, most supervised learning tasks, e.g., the core tasks in computer vision, natural language processing, and speech processing, heavily rely on labeled data. However, labeling data is not a trivial task—it requires educated and knowledgeable annotators (which could be human workers or machine classifiers), to work under a reliable way. More importantly, it needs an effective mechanism to integrate the possibly different labeling from multiple annotators. Techniques addressing this problem in machine learning are called *crowdsourcing* [24] or more generally, *ensemble learning* [8].

---

∗The work is supported in part by the National Science Foundation under projects ECCS 1808159 and NSF ECCS 1608961, and by the Army Research Office (ARO) under projects ARO W911NF-19-1-0247 and ARO W911NF-19-1-0407.

Crowdsourcing has a long history in machine learning, which can be traced back to the 1970s [6]. Many models and methods have appeared since then [22, 23, 21, 34, 38, 28, 37]. Intuitively, if a number of reliable annotators label the same data samples, then *majority voting* among the annotators is expected to work well. However, in practice, not all the annotators are equally reliable—e.g., different annotators could be specialized for recognizing different classes. In addition, not all the annotators are labeling all the data samples, since data samples are often dispatched to different groups of annotators in a certain way. Under such circumstances, majority voting is not very promising.

A more sophisticate way is to treat the crowdsourcing problem as a model identification problem. The arguably most popular generative model in crowdsourcing is the Dawid-Skene model [6], where every annotator is assigned with a 'confusion matrix' that decides the probability of an annotator giving class label $\ell$ when the ground-truth label is $g$. If such confusion matrices and the probability mass function (PMF) of the ground-truth label can be identified, then a maximum likelihood (ML) or a maximum *a posteriori* (MAP) estimator for the true label of any given sample can be constructed. The Dawid-Skene model is quite simple and succinct, and some of the model assumptions (e.g., the conditional independence of the annotator responses) are actually debatable. Nonetheless, this model has been proven very useful in practice [31, 37, 14, 23, 28, 39].

Theoretical aspects for the Dawid-Skene model, however, are less well understood. In particular, it had been unclear if the model could be identified via the accompanying *expectation maximization* (EM) algorithm proposed in the same paper [6], until some recent works addressing certain special cases [23]. The works in [37, 39] put forth tensor methods for learning the Dawid-Skene model. These methods admit model identifiability, and also can be used to effectively initialize the classic EM algorithm provably [39]. The challenge is that tensor methods utilize third-order statistics of the data samples, which are rather hard to estimate reliably in practice given limited data [19].

**Contributions.** In this work, we propose an alternative for identifying the Dawid-Skene model, without using third-order statistics. Our approach is based on utilizing the pairwise co-occurrences of annotators' responses to data samples—which are second-order statistics and thus are naturally much easier to estimate compared to the third-order ones. We show that, by judiciously combining the co-occurrences between different annotator pairs, the confusion matrices and the ground-truth label's prior PMF can be provably identified, under realistic conditions (e.g., when there exists a relatively well-trained annotator among all annotators). This is reminiscent of nonnegative matrix theory and convex geometry [13, 15]. Our approach is also naturally robust to spammers as well as scenarios where every annotator only labels partial data. We offer two algorithms under the same framework. The first algorithm is algebraic, and thus is efficient and suitable for handling very large-scale crowdsourcing problems. The second algorithm offers enhanced identifiability guarantees, and is able to deal with more critical cases (e.g., when no highly reliable annotators exist), with the price of using a computationally more involved iterative optimization algorithm. Experiments show that both approaches outperform a number of competitive baselines.

## 2 Background

**The Dawid-Skene Model.** Let us consider a dataset $\{\boldsymbol{f}_n\}_{n=1}^N$, where $\boldsymbol{f}_n \in \mathbb{R}^d$ is a data sample (or, feature vector) and $N$ is the number of samples. Each $\boldsymbol{f}_n$ belongs to one of $K$ classes. Let $y_n$ be the ground-truth label of the data sample $\boldsymbol{f}_n$. Suppose that there are $M$ annotators who work on the dataset $\{\boldsymbol{f}_n\}_{n=1}^N$ and provide labels. Let $X_m(\boldsymbol{f}_n)$ represent the response of the annotator $m$ to $\boldsymbol{f}_n$. Hence, $X_m$ can be understood as a discrete random variable whose alphabet is $\{1, \dots, K\}$. In crowdsourcing or ensemble learning, our goal is to estimate the true label corresponding to each item $\boldsymbol{f}_n$ from the $M$ annotator responses. Note that in a realistic scenario, an annotator will likely to only work on part of the dataset, since having all annotators work on all the samples is much more costly.

In 1979, Dawid and Skene proposed an intuitively pleasing model for estimating the 'true response' of the patients from recorded answers [6], which is essentially a crowdsourcing/ensemble learning problem. This model has sparked a lot of interest in the machine learning community [31, 37, 14, 23, 28, 39]. The Dawid-Skene model in essence is a *naive Bayesian model* [29]. In this model, the ground-truth label of a data sample is a latent discrete random variable, $Y$, whose values are different class indices. The ambient variables are the responses given by different annotators, denoted as $X_1, \dots, X_M$, where $M$ is the number of annotators. The key assumption in the Dawid-Skene model is that given the ground-truth label, the responses of the annotators are conditionally independent. Of course, the Dawid-Skene model is a simplified version of reality, but has been proven very useful—and it has been a workhorse for crowdsourcing since its proposal.

Under the Dawid-Skene model, one can see that

$$\mathsf{Pr}(X_1 = k_1, \ldots, X_M = k_M) = \sum_{k=1}^{K} \prod_{m=1}^{M} \mathsf{Pr}(X_m = k_m | Y = k)\mathsf{Pr}(Y = k), \tag{1}$$

where $k \in \{1, \ldots, K\}$ denotes the index of a given class, and $k_m$ denotes the response of the $m$-th annotator. If one defines a series of matrices $\boldsymbol{A}_m \in \mathbb{R}^{K \times K}$ and let

$$\boldsymbol{A}(k_m, k) := \mathsf{Pr}(X_m = k_m | Y = k), \tag{2}$$

then $\boldsymbol{A}_m \in \mathbb{R}^{K \times K}$ can be understood as the 'confusion matrix' of annotator $m$: It contains all the conditional probabilities of annotator $m$ labeling a given data sample as from class $k_m$ while the ground-truth label is $k$. Also define a vector $\boldsymbol{d} \in \mathbb{R}^K$ such that $\boldsymbol{d}(k) := \mathsf{Pr}(Y = k)$; i.e., the prior PMF of the ground-truth label $Y$. Then the crowdsourcing problem boils down to estimating $\boldsymbol{A}_m$ for $m = 1, \ldots, M$ and $\boldsymbol{d}$.

**Prior Art.** In the seminal paper [6], Dawid and Skene proposed an EM-based algorithm to estimate $\mathsf{Pr}(X_m = k_m | Y = k)$ and $\mathsf{Pr}(Y = k)$. Their formulation is well-motivated from an ML viewpoint, but also has some challenges. First, it is unknown if the model is identifiable, especially when there is a large number of unrecorded responses (i.e., missing values)—but model identification plays an essential role in such estimation problems [13]. Second, since the ML estimator is a nonconvex optimization criterion, the solution quality of the EM algorithm is not easy to characterize in general. More recently, tensor methods were proposed to identify the Dawid-Skene model [39, 37]. Take the most recent work in [37] as an example. The approach considers estimating the joint probability $\mathsf{Pr}(X_i = k_i, X_j = k_j, X_\ell = k_\ell)$ for different triples $i, j, \ell$. Such joint PMFs can be regarded as third-order tensors, and the confusion matrices and the prior $\boldsymbol{d}$ are latent factors of these tensors. The upshot is that identifiability of $\boldsymbol{A}_m$ and $\boldsymbol{d}$ can be elegantly established leveraging tensor algebra [33, 25]. The challenge, however, is that reliably estimating $\mathsf{Pr}(X_i = k_i, X_j = k_j, X_\ell = k_\ell)$ is quite hard, since it normally needs a large number of annotator responses. Another tensor method in [39] judiciously partitions the data and works with group statistics between three groups, which is reminiscent of the graph statistics proposed in [1]. The method is computationally more tractable, leveraging orthogonal tensor decomposition. Nevertheless, the challenge again lies in sample complexity: the group/graph statistics are still third-order statistics.

## 3 Proposed Approach

In this section, we propose a model identification approach that only uses second-order statistics, in particular, pairwise co-occurrences $\mathsf{Pr}(X_i = k_i, X_j = k_j)$.

**Problem Formulation.** Let us consider the following pairwise joint PMF: $\mathsf{Pr}(X_m = k_m, X_\ell = k_\ell) = \sum_{k=1}^{K} \mathsf{Pr}(Y = k)\mathsf{Pr}(X_m = k_m | Y = k)\mathsf{Pr}(X_\ell = k_\ell | Y = k)$. Letting $\boldsymbol{R}_{m,\ell}(k_m, k_\ell) = \mathsf{Pr}(X_m = k_m, X_\ell = k_\ell)$, and using the matrix notations that we defined, we have $\boldsymbol{R}_{m,\ell}(k_m, k_\ell) = \sum_{k=1}^{K} \mathsf{Pr}(Y = k)\mathsf{Pr}(X_m = k_m | Y = k)\mathsf{Pr}(X_\ell = k_\ell | Y = k)$—or, in a more compact form:

$$\boldsymbol{R}_{m,\ell}(k_m, k_\ell) = \sum_{k=1}^{K} \boldsymbol{d}(k)\boldsymbol{A}_m(k_m, k)\boldsymbol{A}_\ell(k_\ell, k) \iff \boldsymbol{R}_{m,\ell} := \boldsymbol{A}_m \boldsymbol{D} \boldsymbol{A}_\ell^\top,$$

where we have $\boldsymbol{D} = \mathrm{Diag}(\boldsymbol{d})$, which is a diagonal matrix. Note that $\boldsymbol{A}_m$ is a confusion matrix, i.e., its columns are respectable probability measures. In addition, $\boldsymbol{d}$ is a prior PMF. Hence, we have

$$\boldsymbol{1}^\top \boldsymbol{A}_m = \boldsymbol{1}^\top, \ \boldsymbol{A}_m \geq \boldsymbol{0}, \ \forall \, m, \quad \boldsymbol{1}^\top \boldsymbol{d} = 1, \ \boldsymbol{d} \geq \boldsymbol{0}. \tag{3}$$

In practice, $\boldsymbol{R}_{m,\ell}$'s are not available but can be estimated via sample averaging. Specifically, if we are given the annotator responses $X_m(\boldsymbol{f}_n)$, then $\widehat{\boldsymbol{R}}_{m,\ell}(k_m, k_\ell) = \frac{1}{|\mathcal{S}_{m,\ell}|} \sum_{n \in \mathcal{S}_{m,\ell}} I\left[X_m(\boldsymbol{f}_n) = k_m, X_\ell(\boldsymbol{f}_n) = k_\ell\right]$, where $\mathcal{S}_{m,\ell}$ is the index set of samples which both annotators $m$ and $\ell$ have worked on. Here, $I[\cdot]$ is an indicator function: If the event $E$ happens, then $I[E] = 1$, and $I[E^c] = 0$ otherwise. It is readily seen that

$$\mathbb{E}\left[I(X_m(\boldsymbol{f}_n) = k_m, X_\ell(\boldsymbol{f}_n) = k_\ell)\right] = \boldsymbol{R}_{m,\ell}(k_m, k_\ell), \tag{4}$$

where the expectation is taken over data samples. Note that the sample complexity for reliably estimating $\boldsymbol{R}_{m,\ell}$ is much lower relative to that of estimating $\boldsymbol{R}_{m,n,\ell}$ [39, 1], and the latter is needed

in tensor based methods, e.g., [37]. To be specific, to achieve $|\boldsymbol{R}_{m,\ell}(k_m, k_\ell) - \widehat{\boldsymbol{R}}_{m,\ell}(k_m, k_\ell)| \leq \epsilon$ with a probability greater than $1 - \delta$, $\mathcal{O}(\epsilon^{-2}(\log\frac{1}{\delta}))$ joint responses from annotators $m$ and $\ell$ are needed. However, in order to attain the same accuracy for $\widehat{\boldsymbol{R}}_{m,n,\ell}(k_m, k_n, k_\ell)$, the number of joint responses from annotators $m,n$ and $\ell$ is required to be atleast $\mathcal{O}(K\epsilon^{-2}(\log\frac{K}{\delta}))$, where $K$ is the number of classes (also see supplementary materials Sec. J for a short discussion).

**An Algebraic Algorithm.** Assume that we have obtained $\boldsymbol{R}_{m,\ell}$'s for different pairs of $m, \ell$. We now show how to identify $\boldsymbol{A}_m$'s and $\boldsymbol{d}$ from such second-order statistics. Let us take the estimation of $\boldsymbol{A}_m$ as an illustrative example. First, we construct a matrix $\boldsymbol{Z}_m$ as follows:

$$\boldsymbol{Z}_m = \left[ \boldsymbol{R}_{m,m_1}, \boldsymbol{R}_{m,m_2}, \ldots, \boldsymbol{R}_{m,m_{T(m)}} \right], \tag{5}$$

where $m_t \neq m$ for $t = 1, \ldots, T(m)$ denote the indices of annotators who have co-labeled data samples with annotator $m$, and the integer $T(m)$ denotes the number of such annotators. Due to the underlying model of $\boldsymbol{R}_{m,\ell}$ in (3), we have $\boldsymbol{Z}_m = \left[ \boldsymbol{A}_m \boldsymbol{D} \boldsymbol{A}_{m_1}^\top, \ldots, \boldsymbol{A}_m \boldsymbol{D} \boldsymbol{A}_{T(m)}^\top \right] = \boldsymbol{A}_m \left[ \boldsymbol{D} \boldsymbol{A}_{m_1}^\top, \ldots, \boldsymbol{D} \boldsymbol{A}_{T(m)}^\top \right] \in \mathbb{R}^{K \times KT(m)}$. Let us define $\boldsymbol{H}_m^\top = \left[ \boldsymbol{D} \boldsymbol{A}_{m_1}^\top, \ldots, \boldsymbol{D} \boldsymbol{A}_{T(m)}^\top \right] \in \mathbb{R}^{K \times KT(m)}$. This leads to the model $\boldsymbol{Z}_m = \boldsymbol{A}_m \boldsymbol{H}_m^\top$. We propose to identify $\boldsymbol{A}_m$ from $\boldsymbol{Z}_m$. The key enabling postulate is that, among all annotators, some $\boldsymbol{A}_\ell$'s should be *diagonally dominant*—if there exist annotators who are reasonably trained. In other words, for a reasonable annotator $\ell$, $\Pr(X_\ell = j|Y = j)$ should be greater than $\Pr(X_\ell = j|Y = k)$ and $\Pr(X_\ell = j|Y = i)$ for $k, i \neq j$. To see the intuition of the algorithm, consider an ideal case where for each class $k$, there exists an annotator $m_{t(k)} \in \{m_1, \ldots, m_{T(m)}\}$ such that

$$\Pr(X_{m_{t(k)}} = k|Y = k) = 1, \quad \Pr(X_{m_{t(k)}} = k|Y = j) = 0, \quad j \neq k. \tag{6}$$

This physically means that annotator $m_{t(k)}$ is very good at recognizing class $k$ and never confuses other classes with class $k$. Under such circumstances, one can use the following procedure to identify $\boldsymbol{A}_m$. First, let us normalize the columns of $\boldsymbol{Z}_m$ via $\overline{\boldsymbol{Z}}_m(:,q) = \boldsymbol{Z}_m(:,q)/\|\boldsymbol{Z}_m(:,q)\|_1$ for $q = \{1, \ldots, KT(m)\}$. This way, we have a normalized model $\overline{\boldsymbol{Z}}_m = \overline{\boldsymbol{A}}_m \overline{\boldsymbol{H}}_m^\top$, where

$$\overline{\boldsymbol{A}}_m(:,k) = \frac{\boldsymbol{A}_m(:,k)}{\|\boldsymbol{A}_m(:,k)\|_1} = \boldsymbol{A}_m(:,k), \quad \overline{\boldsymbol{H}}_m(q,:) = \frac{\boldsymbol{H}_m(q,:)\|\boldsymbol{A}_m(:,k)\|_1}{\|\boldsymbol{Z}_m(:,q)\|_1}. \tag{7}$$

where the second equality above is because $\|\boldsymbol{A}_m(:,k)\|_1 = 1$ [cf. Eq. (3)]. After normalization, it can be verified that

$$\overline{\boldsymbol{H}}_m \boldsymbol{1} = \boldsymbol{1}, \; \overline{\boldsymbol{H}}_m \geq \boldsymbol{0}, \tag{8}$$

i.e., all the rows of $\overline{\boldsymbol{H}}_m$ reside in the $(K-1)$-probability simplex. In addition, by the assumption in (6), it is readily seen that there exists $\Lambda_q = \{q_1, \ldots, q_K\} \subset \{1, \ldots, L_m\}$ where $L_m = KT(m)$ such that

$$\overline{\boldsymbol{H}}_m(\Lambda_q, :) = \boldsymbol{I}_K, \tag{9}$$

i.e., an identity matrix is a submatrix of $\overline{\boldsymbol{H}}_m$ (after proper row permutations). Consequently, we have $\boldsymbol{A}_m = \overline{\boldsymbol{Z}}_m(:, \Lambda_q)$—i.e., $\boldsymbol{A}_m$ can be identified from $\overline{\boldsymbol{Z}}_m$ up to column permutations. The task also boils down to identifying $\Lambda_q$. This turns out to be a well-studied task in the context of *separable nonnegative matrix factorization* [16, 15, 13], and an algebraic algorithm exists:

$$\widehat{q}_k = \arg \max_{q \in \{1, \ldots, L_m\}} \left\| \boldsymbol{P}_{\widehat{\boldsymbol{A}}_m(:,1:k-1)}^\perp \overline{\boldsymbol{Z}}_m(:,q) \right\|_2^2, \; \forall k. \tag{10}$$

where $\widehat{\boldsymbol{A}}_m(:, 1:k-1) = [\overline{\boldsymbol{Z}}_m(:, \widehat{q}_1), \ldots, \overline{\boldsymbol{Z}}_m(:, \widehat{q}_{k-1})]$ and $\boldsymbol{P}_{\widehat{\boldsymbol{A}}_m(:,1:k-1)}^\perp$ is a projector onto the orthogonal complement of $\text{range}(\widehat{\boldsymbol{A}}_m(:, 1:k-1))$ and we let $\boldsymbol{P}_{\widehat{\boldsymbol{A}}_m(:,1:0)}^\perp := \boldsymbol{I}$.

It has been shown in [16, 2] that the so-called *successive projection algorithm* (SPA) in Eq. (10) identifies $\Lambda_q$ in $K$ steps. This is a very plausible result, since the procedure admits Gram-Schmitt-like lightweight steps and thus is quite scalable. See more details in Sec. F.1.

Each of the $\boldsymbol{A}_m$'s can be estimated from the corresponding $\overline{\boldsymbol{Z}}_m$ by repeatedly applying SPA, and we call this simple procedure *multiple SPA* (`MultiSPA`) as we elaborate in Algorithm 1.

Of course, assuming that (6) or (9) holds perfectly may be too ideal. It is more likely that there exist some annotators who are good at recognizing certain classes, but still have some possibilities of being confused. It is of interest to analyze how SPA can do under such conditions. Another challenge is that one may not have $\boldsymbol{R}_{m,\ell}$ perfectly estimated, since only limited number of samples are available. It is desirable to understand the sample complexity of applying SPA to Dawid-Skene identification. We answer these two key technical questions in the following theorem:

**Theorem 1.** *Assume that annotators $m$ and $t$ co-label at least $S$ samples $\forall t \in \{m_1, \ldots, m_{T(m)}\}$, and that $\widehat{\boldsymbol{Z}}_m$ is constructed*

---

**Algorithm 1** MultiSPA
- **Input:** Annotator Responses $\{X_m(\boldsymbol{f}_n)\}$.
- **Output:** $\widehat{\boldsymbol{A}}_m$ for $m = 1, \ldots, M$, $\widehat{\boldsymbol{d}}$.
- estimate second order statistics $\widehat{\boldsymbol{R}}_{m,\ell}$;
- **for** $m = 1$ **to** $M$ **do**
  - construct $\widehat{\boldsymbol{Z}}_m$ and normalize columns to unit $\ell_1$ norm;
  - estimate $\widehat{\boldsymbol{A}}_m$ using Eq. (10);
- **end for**
- fix permutation mismatch between $\widehat{\boldsymbol{A}}_m$ and $\widehat{\boldsymbol{A}}_\ell$ for all $m \neq \ell$;
- estimate $\widehat{\boldsymbol{D}} = \widehat{\boldsymbol{A}}_m^{-1} \boldsymbol{R}_{m,\ell} (\widehat{\boldsymbol{A}}_\ell^\top)^{-1}$ (and take average over all pairs $(m, \ell)$ if needed).;
- extract the prior $\widehat{\boldsymbol{d}} = \mathrm{diag}(\widehat{\boldsymbol{D}})$.

---

*using $\widehat{\boldsymbol{R}}_{m,m_{T(m)}}$'s according to Eq. (5). Also assume that the constructed $\widehat{\boldsymbol{Z}}_m$ satisfies $\|\widehat{\boldsymbol{Z}}_m(:,l)\|_1 \geq \eta, \forall l \in \{1, \ldots KT(m)\}$, where $\eta \in (0, 1]$. Suppose that $\mathrm{rank}(\boldsymbol{A}_m) = \mathrm{rank}(\boldsymbol{D}) = K$ for $m = 1, \ldots, M$, and that for every class index $k \in \{1, \ldots, K\}$, there exists an annotator $m_{t(k)} \in \{m_1, \ldots, m_{T(m)}\}$ such that*

$$\Pr(X_{m_{t(k)}} = k | Y = k) \geq (1 - \epsilon) \sum_{j=1}^{K} \Pr(X_{m_{t(k)}} = k | Y = j), \tag{11}$$

*where $\epsilon \in [0, 1]$. Then, if $\epsilon \leq \mathcal{O}\left(\max\left(K^{-1}\kappa^{-3}(\boldsymbol{A}_m), \sqrt{\ln(1/\delta)}(\sigma_{\max}(\boldsymbol{A}_m)\sqrt{S}\eta)^{-1}\right)\right)$, with probability greater than $1 - \delta$, the SPA algorithm in (10) can estimate an $\widehat{\boldsymbol{A}}_m$ such that*

$$\left(\min_{\boldsymbol{\Pi}} \|\widehat{\boldsymbol{A}}_m \boldsymbol{\Pi} - \boldsymbol{A}_m\|_{2,\infty}\right) \leq \mathcal{O}\left(\sqrt{K}\kappa^2(\boldsymbol{A}_m) \max\left(\sigma_{\max}(\boldsymbol{A}_m)\epsilon, \sqrt{\ln(1/\delta)}(\sqrt{S}\eta)^{-1}\right)\right) \tag{12}$$

*where $\boldsymbol{\Pi} \in \mathbb{R}^{K \times K}$ is a permutation matrix, $\|\boldsymbol{Y}\|_{2,\infty} = \max_\ell \|\boldsymbol{Y}(:, \ell)\|_2$, $\sigma_{\max}(\boldsymbol{A}_m)$ is the largest singular value of $\boldsymbol{A}_m$, and $\kappa(\boldsymbol{A}_m)$ is the condition number of $\boldsymbol{A}_m$.*

In the above Theorem, the assumption $\|\widehat{\boldsymbol{Z}}_m(:, l)\|_1 \geq \eta$ means that the proposed algorithm favors cases where more co-occurrences are observed, since $\widehat{\boldsymbol{Z}}_m$'s elements are averaged number of co-occurrences—which makes a lot of sense. In addition, Eq. (11) relaxes the ideal assumption in (6), allowing the 'good annotator' $m_{t(k)}$ to confuse class $j \neq k$ with class $k$ up to a certain probability, thereby being more realistic. The proof of Theorem 1 is reminiscent of the noise robustness of the SPA algorithm [16, 2]; see the supplementary materials (Sec. F.1). A direct corollary is as follows:

**Corollary 1.** *Assume that the conditions in Theorem 1 hold for $\widehat{\boldsymbol{Z}}_m$ and $\boldsymbol{A}_m$, $\forall m \in \{1, \ldots, M\}$. Then, the estimation error bound in (12) holds for every MultiSPA-output $\widehat{\boldsymbol{A}}_m$, $\forall m \in \{1, \ldots, M\}$.*

Theorem 1 and Corollary 1 are not entirely surprising due to the extensive research on SPA-like algorithms [2, 16, 10, 30, 4]. The implication for crowdsourcing, however, is quite intriguing. First, one can see that if an annotator $m$ does not label all the data samples, it does not necessarily hurt the model identifiability—as long as annotator $m$ has co-labeled some samples with a number of other annotators, identification of $\boldsymbol{A}_m$ is possible. Second, assume that there exists a well-trained annotator $m^\star$ whose confusion matrix is diagonally dominant, then for every annotator $m$ who has co-labeled samples with annotator $m^\star$, the matrix $\overline{\boldsymbol{H}}_m$ can easily satisfy (11) by letting $m_{t(k)} = m^\star$ for all $k$. In practice, one would not know who is $m^\star$—otherwise the crowdsourcing problem would be trivial. However, one can design a dispatch strategy such that every pair of annotators $m$ and $\ell$ co-label a certain amount of data. This way, it guarantees that $\boldsymbol{A}_{m^\star}$ appears in everyone else's $\boldsymbol{H}_m$ and thus ensures identifiability of all $\boldsymbol{A}_m$'s for $m \neq m^\star$. This insight may shed some light on how to effectively dispatch data to annotators.

Another interesting question to ask is *does having more annotators help?* Intuitively, having more annotators should help: If one has more rows in $\overline{\boldsymbol{H}}_m$, then it is more likely that some rows approach

the vertices of the probability simplex—which can then enable SPA. We use the following simplified generative model and theorem to formalize the intuition:

**Theorem 2.** *Let $\rho > 0, \varepsilon > 0$, and assume that the rows of $\overline{\boldsymbol{H}}_m$ are generated within the $(K-1)$-probability simplex uniformly at random. If the number of annotators satisfies $M \geq \Omega\left(\frac{\varepsilon^{-2(K-1)}}{K}\log\left(\frac{K}{\rho}\right)\right)$, then, with probability greater than or equal to $1 - \rho$, there exist rows of $\overline{\boldsymbol{H}}_m$ indexed by $q_1, \ldots q_K$ such that $\|\overline{\boldsymbol{H}}_m(q_k, :) - \boldsymbol{e}_k^\top\|_2 \leq \varepsilon, \ k = 1, \ldots, K$.*

Note that Theorem 2 implies (11) under proper $\varepsilon$ and $\epsilon$—and thus having more annotators indeed helps identify the model. The above can be shown by utilizing the Chernoff-Hoeffding inequality, and the detailed proof can be found in the supplementary materials (Sec. G).

After obtaining $\widehat{\boldsymbol{A}}_m$'s, $\boldsymbol{d}$ can be estimated via various ways—see the supplementary materials in Sec. D. Using $\widehat{\boldsymbol{d}}$ and $\widehat{\boldsymbol{A}}_m$'s together, ML and MAP estimators for the true labels can be built up [37].

## 4  Identifiability-enhanced Algorithm

The `MultiSPA` algorithm is intuitive and lightweight, and is effective as we will show in the experiments. One concern is that perhaps the assumption in (11) may be violated in some cases. In this section, we propose another model identification algorithm that is potentially more robust to critical scenarios. Specifically, we consider the following feasibility problem:

$$
\begin{aligned}
\text{find} \quad & \{\boldsymbol{A}_m\}_{m=1}^M, \ \boldsymbol{D} && \text{(13a)}\\
\text{subject to} \quad & \boldsymbol{R}_{m,\ell} = \boldsymbol{A}_m \boldsymbol{D} \boldsymbol{A}_\ell^\top, \ \forall m, \ell \in \{1, \ldots, M\} && \text{(13b)}\\
& \boldsymbol{1}^\top \boldsymbol{A}_m = \boldsymbol{1}^\top, \ \boldsymbol{A}_m \geq \boldsymbol{0}, \ \forall m, \ \boldsymbol{1}^\top \boldsymbol{d} = 1, \ \boldsymbol{d} \geq \boldsymbol{0}. && \text{(13c)}
\end{aligned}
$$

The criterion in (13) seeks confusion matrices and a prior PMF that fit the available second-order statistics. The constraints in (13c) reflect the fact that the columns of $\boldsymbol{A}_m$'s are conditional PMFs and the prior $\boldsymbol{d}$ is also a PMF.

To proceed, let us first introduce the following notion from convex geometry [13, 27]:

**Definition 1.** *(Sufficiently Scattered) A nonnegative matrix $\boldsymbol{H} \in \mathbb{R}^{L \times K}$ is sufficiently scattered if 1) $\mathrm{cone}\{\boldsymbol{H}^\top\} \supseteq \mathcal{C}$, and 2) $\mathrm{cone}\{\boldsymbol{H}^\top\}^* \cap \mathrm{bd}\,\mathcal{C}^* = \{\lambda \boldsymbol{e}_k \mid \lambda \geq 0, k = 1, ..., K\}$. Here, $\mathcal{C} = \{\boldsymbol{x} | \boldsymbol{x}^\top \boldsymbol{1} \geq \sqrt{K-1}\|\boldsymbol{x}\|_2\}$, $\mathcal{C}^* = \{\boldsymbol{x} | \boldsymbol{x}^\top \boldsymbol{1} \geq \|\boldsymbol{x}\|_2\}$. In addition, $\mathrm{cone}\{\boldsymbol{H}^\top\} = \{\boldsymbol{x} | \boldsymbol{x} = \boldsymbol{H}^\top \boldsymbol{\theta}, \ \forall \boldsymbol{\theta} \geq \boldsymbol{0}\}$ and $\mathrm{cone}\{\boldsymbol{H}^\top\}^* = \{\boldsymbol{y} | \boldsymbol{x}^\top \boldsymbol{y} \geq 0, \ \forall \boldsymbol{x} \in \mathrm{cone}\{\boldsymbol{H}^\top\}\}$ are the conic hull of $\boldsymbol{H}^\top$ and its dual cone, respectively, and $\mathrm{bd}$ is the boundary of a closed set.*

The sufficiently scattered condition has recently emerged in convex geometry-based matrix factorization [27, 12]. This condition models how the rows of $\boldsymbol{H}$ are spread in the nonnegative orthant. In principle, the sufficiently scattered condition is much easier to be satisfied relative to the condition as in (9), or, the so-called *separability condition* under the context of nonnegative matrix factorization [9, 16]. $\boldsymbol{H}$ satisfying the separability condition is the extreme case, meaning that $\mathrm{cone}\{\boldsymbol{H}^\top\} = \mathbb{R}_+^K$. However, the sufficiently scattered condition only requires $\mathcal{C} \subseteq \mathrm{cone}\{\boldsymbol{H}^\top\}$—which is naturally much more relaxed; also see [13] and the supplementary materials for detailed illustrations (Sec. E).

Regarding identifiability of $\boldsymbol{A}_1, \ldots, \boldsymbol{A}_M$ and $\boldsymbol{d}$, we have the following result:

**Theorem 3.** *Assume that $\mathrm{rank}(\boldsymbol{D}) = \mathrm{rank}(\boldsymbol{A}_m) = K$ for all $m = 1, \ldots, M$, and that there exist two subsets of the annotators, indexed by $\mathcal{P}_1$ and $\mathcal{P}_2$, where $\mathcal{P}_1 \cap \mathcal{P}_2 = \emptyset$ and $\mathcal{P}_1 \cup \mathcal{P}_2 \subseteq \{1, \ldots, M\}$. Suppose that from $\mathcal{P}_1$ and $\mathcal{P}_2$ the following two matrices can be constructed: $\boldsymbol{H}^{(1)} = [\boldsymbol{A}_{m_1}^\top, \ldots, \boldsymbol{A}_{m_{|\mathcal{P}_1|}}^\top]^\top$, $\boldsymbol{H}^{(2)} = [\boldsymbol{A}_{\ell_1}^\top, \ldots, \boldsymbol{A}_{\ell_{|\mathcal{P}_2|}}^\top]^\top$, where $m_t \in \mathcal{P}_1$ and $\ell_j \in \mathcal{P}_2$. Furthermore, assume that i) both $\boldsymbol{H}^{(1)}$ and $\boldsymbol{H}^{(2)}$ are sufficiently scattered; ii) all $\boldsymbol{R}_{m_t, \ell_j}$'s for $m_t \in \mathcal{P}_1$ and $\ell_j \in \mathcal{P}_2$ are available; and iii) for every $m \notin \mathcal{P}_1 \cup \mathcal{P}_2$ there exists a $\boldsymbol{R}_{m,r}$ available, where $r \in \mathcal{P}_1 \cup \mathcal{P}_2$. Then, solving Problem (13) recovers $\boldsymbol{A}_m$ for $m = 1, \ldots, M$ and $\boldsymbol{D} = \mathrm{Diag}(\boldsymbol{d})$ up to identical column permutation.*

The proof of Theorem 3 is relegated to the supplementary results (Sec. H). Note that the theorem holds under the the existence of $\mathcal{P}_1$ and $\mathcal{P}_2$, but there is no need to know the sets *a priori*. Generally

speaking, a 'taller' matrix $\boldsymbol{H}^{(i)}$ would have a better chance to have its rows sufficiently spread in the nonnegative orthant under the same intuition of Theorem 2. Thus, having more annotators also helps to attain the sufficiently scattered condition. Nevertheless, formally showing the relationship between the number of annotators and $\boldsymbol{H}^{(i)}$ for $i = 1, 2$ being sufficiently scattered is more challenging than the case in Theorem 2, since the sufficiently scattered condition is a bit more abstract relative to the separability condition—the latter specifically assumes $\boldsymbol{e}_k$'s exist as rows of $\boldsymbol{H}^{(i)}$ while the former depends on the 'shape' of the conic hull of $(\boldsymbol{H}^{(i)})^\top$, which contains an infinite number of cases. Towards this end, let us first define the following notion:

**Definition 2.** *Assume that there exist $\widetilde{\boldsymbol{H}} \in \mathbb{R}^{L \times K}$ such that $\widetilde{\boldsymbol{H}}$ is sufficiently scattered. Also assume $\mathcal{V}$ is the row index set of $\widetilde{\boldsymbol{H}}$ such that $\widetilde{\boldsymbol{H}}(\mathcal{V}, :)$ collects the extreme rays of $\mathsf{cone}\{\widetilde{\boldsymbol{H}}^\top\}$. If there exist row indices $\ell_v \in \{1, \ldots, L\}$ for all $v \in \mathcal{V}$, such that $\|\widetilde{\boldsymbol{H}}(v, :) - \boldsymbol{H}(\ell_v, :)\|_2 \leq \varepsilon$, then $\boldsymbol{H} \in \mathbb{R}^{L \times K}$ is called $\varepsilon$-sufficiently scattered.*

One can see that an $\varepsilon$-sufficiently scattered matrix is sufficiently scattered when $\varepsilon \to 0$. With this definition, we show the following theorem:

**Theorem 4.** *Let $\rho > 0, \frac{\alpha}{2} > \varepsilon > 0$,, and assume that the rows of $\boldsymbol{H}^{(1)}$ and $\boldsymbol{H}^{(2)}$ are generated from $\mathbb{R}^K$ uniformly at random. If the number of annotators satisfies $M \geq \Omega\left(\frac{(K-1)^2}{K\alpha^{2(K-2)}\varepsilon^2}\log\left(\frac{K(K-1)}{\rho}\right)\right)$, where $\alpha = 1$ for $K = 2$, $\alpha = 2/3$ for $K = 3$ and $\alpha = 1/2$ for $K > 3$, then with probability greater than or equal to $1 - \rho$, $\boldsymbol{H}^{(1)}$ and $\boldsymbol{H}^{(2)}$ are $\varepsilon$-sufficiently scattered.*

The proof of Theorem 4 is relegated to the supplementary materials (Sec. I). One can see that to satisfy $\varepsilon$-sufficiently scattered condition, $M$ is smaller than that in Theorem 2. Conditions *i)-iii)* in Theorem 3 and Theorem 4 together imply that if we have enough annotators, and if many pairs co-label a certain number of data, then it is quite possible that one can identify the Dawid-Skene model via simply finding a feasible solution to (13). This feasibility problem is nonconvex, but can be effectively approximated; see the supplementary materials (Sec. C). In a nutshell, we reformulate the problem as a Kullback-Leibler (KL) divergence-based constrained fitting problem and handle it using alternating optimization. Since nonconvex optimization relies on initialization heavily, we use `MultiSPA` to initialize the fitting stage—which we will refer to as the `MultiSPA-KL` algorithm.

## 5  Experiments

**Baselines.** The performance of the proposed approach is compared with a number of competitive baselines, namely, `Spectral-D&S` [39], `TensorADMM` [37], and `KOS` [22], `EigRatio` [5], `GhoshSVD` [14] and `MinmaxEntropy` [40]. The performance of the `Majority Voting` scheme and the Majority Voting initialized Dawid-Skene (`MV-D&S`) estimator [6] are also presented. We also use `MultiSPA` to initialize EM algorithm (named as `MultiSPA-D&S`). Note that `KOS`, `EigRatio` and `MinmaxEntropy` work with more complex models relative to the Dawid-Skene model, but are considered as good baselines for the crowdsourcing/ensemble learning tasks. After identifying the model parameters, we construct a MAP predictor following [37] and observe the result. The algorithms are coded in Matlab.

**Synthetic-data Simulations.** Due to page limitations, synthetic data experiments demonstrating model identifiability of the proposed algorithms are presented in the supplementary materials (Sec. A).

**Integrating Machine Classifiers.** We employ different UCI datasets (`https://archive.ics.uci.edu/ml/datasets.html`; details in Sec. B). For each of the datasets under test, we use a collection of different classification algorithms to annotate the data samples. Different classification algorithms from the MATLAB machine learning toolbox (`https://www.mathworks.com/products/statistics.html`) such as various $k$-nearest neighbour classifiers, support vector machine classifiers, and decision tree classifiers are employed to serve as our machine annotators. In order to train the annotators, we use $20\%$ of the samples to act as training data. After the data samples are trained, we use the annotators to label the unseen data samples. In practice, not all samples are labeled by an annotator due to several factors such as annotator capacity, difficulty of the task, economical issues and so on. To simulate such a scenario, each of the trained algorithms is allowed to label a data sample with probability $p \in (0, 1]$. We test the performance of all the algorithms under different $p$'s—and a smaller $p$ means a more challenging scenario. All the results are averaged from 10 random trials.

Table 1 shows the classification error of the algorithms under test. Since `GhoshSVD` and `EigenRatio` works only on binary tasks, they are not evaluated for the Nursery dataset where $K = 4$. The 'single

Table 1: Classification Error (%) on UCI Datasets; see runtime tabulated in Sec. B.

| | Nursery | | | Mushroom | | | Adult | | |
|---|---|---|---|---|---|---|---|---|---|
| **Algorithms** | $p=1$ | $p=0.5$ | $p=0.2$ | $p=1$ | $p=0.5$ | $p=0.2$ | $p=1$ | $p=0.5$ | $p=0.2$ |
| `MultiSPA` | 2.83 | 4.54 | 17.96 | 0.02 | 0.293 | 6.35 | **15.71** | **16.05** | 17.66 |
| `MultiSPA-KL` | **2.72** | **4.26** | **13.06** | **0.00** | **0.152** | **5.89** | 15.66 | 15.98 | 17.63 |
| `MultiSPA-D&S` | **2.82** | **4.44** | 13.39 | **0.00** | 0.194 | 6.17 | 15.74 | 16.29 | 23.88 |
| `Spectral-D&S` | 3.14 | 37.2 | 44.29 | **0.00** | 0.198 | 6.17 | 15.72 | 16.31 | 23.97 |
| `TensorADMM` | 17.97 | 7.26 | 19.78 | 0.06 | 0.237 | 6.18 | 15.72 | **16.05** | 25.08 |
| `MV-D&S` | 2.92 | 66.48 | 66.61 | **0.00** | 47.99 | 48.63 | 15.76 | 75.21 | 75.13 |
| `Minmax-entropy` | 3.63 | 26.31 | **11.09** | **0.00** | **0.163** | 8.14 | 16.11 | 16.92 | **15.64** |
| `EigenRatio` | N/A | N/A | N/A | 0.06 | 0.329 | **5.97** | 15.84 | 16.28 | 17.69 |
| `KOS` | 4.21 | 6.07 | 13.48 | 0.06 | 0.576 | 6.42 | 17.19 | 24.97 | 38.29 |
| `Ghosh-SVD` | N/A | N/A | N/A | 0.06 | 0.329 | **5.97** | 15.84 | 16.28 | 17.71 |
| `Majority Voting` | 2.94 | 4.83 | 19.75 | 0.14 | 0.566 | 6.57 | 15.75 | 16.21 | 20.57 |
| Single Best | 3.94 | N/A | N/A | 0.00 | N/A | N/A | 16.23 | N/A | N/A |
| Single Worst | 15.65 | N/A | N/A | 7.22 | N/A | N/A | 19.27 | N/A | N/A |

Table 2: Classification Error (%) and Run-time (sec) : AMT Datasets

| Algorithms | TREC | | Bluebird | | RTE | | Web | | Dog | |
|---|---|---|---|---|---|---|---|---|---|---|
| | (%) Error | (sec) Time | (%) Error | (sec) Time | (%) Error | (sec) Time | (%) Error | (sec) Time | (%) Error | (sec) Time |
| `MultiSPA` | 31.47 | 50.68 | 13.88 | 0.07 | 8.75 | 0.28 | 15.22 | 0.54 | 17.09 | 0.07 |
| `MultiSPA-KL` | **29.23** | 536.89 | **11.11** | 1.94 | **7.12** | 17.06 | **14.58** | 12.34 | **15.48** | 15.88 |
| `MultiSPA-D&S` | 29.84 | 53.14 | 12.03 | 0.09 | **7.12** | 0.32 | 15.11 | 0.84 | **16.11** | 0.12 |
| `Spectral-D&S` | **29.58** | 919.98 | 12.03 | 1.97 | **7.12** | 6.40 | 16.88 | 179.92 | 17.84 | 51.16 |
| `TensorADMM` | N/A | N/A | 12.03 | 2.74 | N/A | N/A | N/A | N/A | 17.96 | 603.93 |
| `MV-D&S` | 30.02 | 3.20 | 12.03 | 0.02 | 7.25 | 0.07 | 16.02 | 0.28 | 15.86 | 0.04 |
| `Minmax-entropy` | 91.61 | 352.36 | **8.33** | 3.43 | 7.50 | 9.10 | **11.51** | 26.61 | 16.23 | 7.22 |
| `EigenRatio` | 43.95 | 1.48 | 27.77 | 0.02 | 9.01 | 0.03 | N/A | N/A | N/A | N/A |
| `KOS` | 51.95 | 9.98 | **11.11** | 0.01 | 39.75 | 0.03 | 42.93 | 0.31 | 31.84 | 0.13 |
| `GhoshSVD` | 43.03 | 11.62 | 27.77 | 0.01 | 49.12 | 0.03 | N/A | N/A | N/A | N/A |
| `Majority Voting` | 34.85 | N/A | 21.29 | N/A | 10.31 | N/A | 26.93 | N/A | 17.91 | N/A |

best' and 'single worst' rows correspond to the results of using the classifiers individually when $p = 1$, as references. The best and second-best performing algorithms are highlighted in the table. One can see that the proposed methods are quite promising for this experiment. Both algorithms largely outperform the tensor based methods `TensorADMM` and `Spectral-D&S` in this case, perhaps because the limited number of available samples makes the third-order statistics hard to estimate. It is also observed that the proposed algorithms enjoy favorable runtime;s ee supplementary materials (cf. Table 8 in Sec. B). Using the `MultiSPA` to initialize EM (i.e. `MultiSPA-D&S`) also works well, which offers another viable option that strikes a good balance between runtime and accuracy.

**Amazon Mechanical Turk Crowdsourcing Data.** In this section, the performance of the proposed algorithms are evaluated using the Amazon Mechanical Turk (AMT) data (`https://www.mturk.com`) in which human annotators label various classification tasks. Data description is given in the supplementary materials Sec. B. Table 2 shows the classification error and the runtime performance of the algorithms under test. One can see that `MultiSPA` has a very favorable execution time, because it is a Gram-Schmitt-like algorithm. `MultiSPA-KL` uses more time, because it is an iterative optimization method—with better accuracy paid off. Since `TensorADMM` algorithm does not scale well, the results are not reported for very large datasets (i.e., TREC and RTE). Similar as before, since Web and Dog are multi-class datasets, `EigenRatio` and `GhoshSVD` are not applicable. From the results, it can be seen that the proposed algorithms outperform many existing crowdsourcing algorithms in both classification accuracy and runtime. In particular, one can see that the algebraic algorithm `MultiSPA` gives very similar results compared to the computationally much more involved algorithms. This shows the potential for its application in big data crowdsourcing.

## 6 Conclusion

In this work, we have revisited the classic Dawid-Skene model for multi-class crowdsourcing. We have proposed a second-order statistics-based approach that guarantees identifiability of the model parameters, i.e., the confusion matrices of the annotators and the label prior. The proposed method naturally admits lower sample complexity relative to existing methods that utilize tensor algebra to ensure model identifiability. The proposed approach also has an array of favorable features. In particular, our framework enables a lightweight algebraic algorithm, which is reminiscent of the Gram-Schmitt-like SPA algorithm for nonnegative matrix factorization. We have also proposed a coupled and constrained matrix factorization criterion that enjoys enhanced-identifiability, as well as an alternating optimization algorithm for handling the identification problem. Real-data experiments show that our proposed algorithms are quite promising for integrating crowdsourced labeling.

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
