[Supplementary Material · neurips_2019_supp_after_acceptance.pdf]

**Supplementary Materials for "Crowdsourcing via Pairwise Co-occurrences: Identifiability and Algorithms"**

## A   Synthetic Data Experiments

In the first experiment, we consider that $M = 25$ annotators are available to annotate $N = 10,000$ items, each belonging to one of $K = 3$ classes. The true label for each item is sampled uniformly from $\{1, \ldots, K\}$, i.e, the prior probability vector $\boldsymbol{d}$ is fixed to be $\boldsymbol{d} = [1/3, 1/3, 1/3]^\top$. For generating the confusion matrices, two different cases are considered

- **Case 1**: an annotator is chosen uniformly at random and is assigned an ideal confusion matrix, ie., an identity matrix $\boldsymbol{I}_3$. This ensures the assumption as given by Eq.(9) (or Eq. (6)).

- **Case 2**: an annotator $m$ is chosen uniformly at random and its confusion matrix is made diagonally dominant such that $\boldsymbol{A}_m(k, k) > \boldsymbol{A}_m(k', k)$, for $k', k \in \{1, \ldots, K\}, k \neq k'$. To achieve this, the elements of each column of $\boldsymbol{A}_m$ is drawn from a uniform distribution between 0 and 1. The columns are then normalized using their respective $\ell_1$-norms. After that, for each column, the elements are re-organized such that the corresponding diagonal entry is dominant in that column and then normalized with respect to $\ell_1$-norm. In this way, Eq. (11) in Theorem 1 may be (approximately) satisfied.

In both the cases, for the remaining annotators, the confusion matrices $\boldsymbol{A}_m$ are randomly generated; the elements are first drawn following the uniform distribution between 0 and 1, and then the columns are normalized with respect to the $\ell_1$-norm. Once $\boldsymbol{A}_m$'s are generated, the responses from each annotator $m$ for the items with true labels $g \in \{1, \ldots, K\}$ are randomly chosen from $\{1, \ldots, K\}$ using the probability distribution $\boldsymbol{A}_m(:, g)$. An annotator response for each item is retained for the estimation of $\boldsymbol{A}_m$ with probability $p \in (0, 1]$. In other words, with probability $1 - p$, each response is made 0. In this way, our simulated scenario is expected to mimic realistic situations where we have a combination of reliable and unreliable annotators, each labeling parts of the items. Using the generated responses, we construct $\widehat{\boldsymbol{R}}_{m,\ell}$'s and then follow the proposed approach to identify the confusion matrices and the prior $\boldsymbol{d}$.

The accuracy of the estimation is measured using *mean squared error* (MSE) defined as,

$$\text{MSE} = \frac{1}{M} \sum_{m=1}^{M} \text{MSE}_m, \tag{14}$$

where,

$$\text{MSE}_m = \min_{\pi(k) \in \{1, \ldots, K\}} \frac{1}{K} \sum_{k=1}^{K} \|\boldsymbol{A}_m(:, \pi(k)) - \widehat{\boldsymbol{A}}_m(:, k)\|_2^2 \tag{15}$$

where $\widehat{\boldsymbol{A}}_m$ is the estimate of $\boldsymbol{A}_m$ and $\pi(k)$'s are used to fix the column permutation.

The average (MSE) of the confusion matrices for various values of $p$ under the above mentioned cases are shown in Table 3 and Table 4 where the proposed methods, `MultiSPA` and `MultiSPA-KL` are compared with the baselines `Spectral-E&M`, `TensorADMM` and `MV-D&S` since these methods are also Dawid-Skene model identification approaches. As `MV-D&S` becomes numerically unstable for smaller values of $p$, those results are not reported in the table. All the results are averaged from 10 trials.

From the two tables, one can see that `MultiSPA` works reasonably well for both cases. As expected, it exhibits lower MSEs for case 1, since the condition in (6) is perfectly enforced. Nevertheless, in both cases, using `MultiSPA` to initialize the `KL` algorithm identifies the confusion matrices to a very high accuracy. It is observed that `MultiSPA-KL` outperforms the baselines in terms of the estimation accuracy —which may be a result of using second order statistics.

Under the same settings as in case 2, the true labels are estimated using the MAP/ML predictor as in [37] (in this case, ML and MAP are the same since the prior PMF is a uniform distribution). The classification error and the runtime of the crowdsourcing algorithms are computed and shown in Table 5.

Table 3: Average MSE of the confusion matrices $\boldsymbol{A}_m$ for case 1.

| Algorithms | $p = 0.2$ | $p = 0.3$ | $p = 0.5$ | $p = 1$ |
|---|---|---|---|---|
| MutliSPA | 0.0184 | 0.0083 | 0.0063 | 0.0034 |
| MultiSPA-KL | **0.0019** | **0.0009** | **0.0004** | **1.73E-04** |
| Spectral D&S | 0.0320 | 0.0112 | 0.0448 | 1.74E-04 |
| TensorADMM | 0.0026 | 0.0011 | 0.0005 | 1.88E-04 |
| MV-D&S | – | – | 0.0173 | 1.84E-04 |

Table 4: Average MSE of the confusion matrices $\boldsymbol{A}_m$ for case 2.

| Algorithms | $p = 0.2$ | $p = 0.3$ | $p = 0.5$ | $p = 1$ |
|---|---|---|---|---|
| MutliSPA | 0.0229 | 0.0188 | 0.0115 | 0.0102 |
| MultiSPA-KL | **0.0029** | **0.0014** | **0.0005** | 1.67E-04 |
| Spectral D&S | 0.0348 | 0.0265 | 0.0391 | **1.67E-04** |
| TensorADMM | 0.0031 | 0.0016 | 0.0006 | 1.93E-04 |
| MV-D&S | – | – | 0.0028 | 5.88E-04 |

In the next experiment with case 2, the true labels are sampled with unequal probability. Specifically, $\boldsymbol{d}$ is set to be $[\frac{1}{6}, \frac{2}{3}, \frac{1}{6}]^\top$ with all other parameters and conditions same as in the first experiment. Using the MAP predictor, the true labels are estimated for the proposed algorithms for various values of $p$ and the results are shown in Table 6. It can be inferred from the results that both the proposed algorithms MultiSPA and MultiSPA-KL grantee better classification accuracy when the true label distribution of the items is not balanced.

In the next experiment, the effect of the number of annotators $(M)$ in the estimation accuracy of the confusion matrices is investigated. According to Theorem 2 and 4, the proposed methods will benefit from the availability of more annotators (i.e., a larger $M$). For $N = 10,000$, $K = 3$, $\boldsymbol{d} = [\frac{1}{6}, \frac{2}{3}, \frac{1}{6}]^\top$, $p = 0.5$ and the true confusion matrices $\boldsymbol{A}_m$ being generated as in case 2, the MSEs under various values of $M$ are plotted in Figure 1. One can see that MultiSPA-KL achieves better accuracy relative to MultiSPA under the same $M$'s, which corroborates our results in Theorem 4.

Figure 1: MSE of the confusion matrices for various values of $M$

Table 5: Classification Error(%) & Averge run-time when $\boldsymbol{d} = [\frac{1}{3}, \frac{1}{3}, \frac{1}{3}]^\top$

| Algorithms | $p = 0.2$ | $p = 0.3$ | $p = 0.5$ | Run-time(sec) |
|---|---|---|---|---|
| MultiSPA | 37.24 | 26.39 | 19.21 | 0.049 |
| MultiSPA-KL | **31.71** | **21.10** | **12.79** | 18.07 |
| MultiSPA-D&S | **31.95** | **21.11** | **12.80** | 0.069 |
| Spectral-D&S | 46.37 | 23.92 | 12.89 | 27.17 |
| TensorADMM | 32.16 | 21.34 | 12.91 | 56.09 |
| MV-D&S | 66.91 | 57.92 | 13.09 | 0.096 |
| Minmax-entropy | 62.83 | 65.50 | 67.31 | 200.91 |
| KOS | 71.47 | 61.05 | 13.12 | 5.653 |
| Majority Voting | 67.57 | 68.37 | 71.39 | – |

Table 6: Classification Error(%) & Averge run-time when $\boldsymbol{d} = [\frac{1}{6}, \frac{2}{3}, \frac{1}{6}]^\top$

| Algorithms | $p = 0.2$ | $p = 0.3$ | $p = 0.5$ | Run-time(sec) |
|---|---|---|---|---|
| MultiSPA | **30.75** | **21.29** | **13.67** | 0.105 |
| MultiSPA-KL | **23.19** | **16.62** | **10.13** | 18.93 |
| MultiSPA-D&S | 40.12 | 32.1 | 21.46 | 0.122 |
| Spectral-D&S | 56.17 | 49.41 | 39.17 | 28.01 |
| TensorADMM | 34.17 | 25.53 | 11.97 | 152.76 |
| MV-D&S | 83.14 | 83.15 | 32.98 | 0.090 |
| Minmax-entropy | 83.04 | 63.08 | 74.29 | 232.82 |
| KOS | 70.79 | 67.55 | 78.00 | 6.19 |
| Majority Voting | 65.37 | 65.57 | 66.06 | – |

## B  More Details on UCI and AMT Dataset Experiments

**UCI data.** The details of the UCI datasets employed in the real data experimemts is given in Table 7. To be more specific, the Adult dataset predicts the income of a person into $K = 2$ classes based on 14 attributes. The Mushroom dataset has 22 attributes of certain variations of mushrooms and the task there predicts either 'edible' or 'poisonous'. The Nursery dataset predicts applications to one of the 4 categories based on 8 attributes of the financial and social status of the parents.

The proposed methods and the baselines are compared in terms of runtime for various datasets and the results are reported in Table 8. All the results are averaged from 10 different trials.

**AMT data.** The Amazon Mechanical Turk (AMT) datasets used in our crowdsourcing data experiments is given in Table 9. Specifically, the tasks involving the Bird dataset [38], the RTE dataset [34], and the TREC dataset [26], are binary classification tasks. The tasks associated with the Dog dataset [7] and the web dataset [40] are multi-class tasks (i.e., 4 and 5 classes, respectively).

We would like to add one remark regarding the two-stage approaches that involving an initial stage and a refinement stage (e.g., Spectral-D&S, MV-D&S, and MultiSPA-KL). Due to very high sparsity of the annotator responses in most of the AMT data, the estimated confusion matrices from the first stage may contain many zero entries, which may sometimes lead to numerical issues in the second stage, as observed in [39]. In our experiments, we follow an empirical thresholding strategy proposed in [39]. Specifically, the confusion matrix entries that are smaller than a threshold $\Delta$ are reset to $\Delta$ and the columns are normalized before initialization. In our experiments, we use $\Delta = 10^{-6}$ for most of the cases except the extremely large dataset TREC, which enjoys better performance of all methods using $\Delta = 10^{-5}$.

Table 7: Details of UCI Datasets.

| UCI dataset name | # classes | # items | # annotators |
|---|---|---|---|
| Adult | 2 | 7017 | 10 |
| Mushroom | 2 | 6358 | 10 |
| Nursery | 4 | 3575 | 10 |

Table 8: Average runtime (sec) for UCI datset experiments.

| Algorithms | Nursery | Mushroom | Adult |
|---|---|---|---|
| MultiSPA | 0.021 | 0.012 | 0.018 |
| MultiSPA-KL | 1.112 | 0.663 | 0.948 |
| MultiSPA-D&S | 0.035 | 0.027 | 0.027 |
| Spectral-D&S | 10.09 | 0.496 | 0.512 |
| TensorADMM | 5.811 | 0.743 | 4.234 |
| MV-D&S | 0.009 | 0.007 | 0.008 |
| Minmax-entropy | 19.94 | 2.304 | 6.959 |
| EigenRatio | – | 0.005 | 0.007 |
| KOS | 0.768 | 0.085 | 0.118 |
| Ghosh-SVD | – | 0.081 | 0.115 |

Table 9: AMT Dataset description.

| Dataset | # classes | # items | # annotators | # annotator labels |
|---|---|---|---|---|
| Bird | 2 | 108 | 30 | 3240 |
| RTE | 2 | 800 | 164 | 8,000 |
| TREC | 2 | 19,033 | 762 | 88,385 |
| Dog | 4 | 807 | 52 | 7,354 |
| Web | 5 | 2,665 | 177 | 15,567 |

## C  Algorithm for Criterion (13)

In this section, the `MultiSPA-KL` algorithm is discussed in detail. To implement the identification criterion in (13), we lift the constraint (13b) and employ the following coupled matrix factorization cirterion:

$$\underset{\{\boldsymbol{A}_m\}_{m=1}^M,\ \boldsymbol{D}}{\text{minimize}} \sum_{m,\ell} \mathsf{KL}\left(\widehat{\boldsymbol{R}}_{m,\ell}||\boldsymbol{A}_m\boldsymbol{D}\boldsymbol{A}_\ell^\top\right), \tag{16a}$$

$$\text{subject to}: \mathbf{1}^\top\boldsymbol{A}_m = \mathbf{1}^\top,\ \boldsymbol{A}_m \geq \mathbf{0},\ \mathbf{1}^\top\boldsymbol{d} = 1,\ \boldsymbol{d} \geq \mathbf{0}, \tag{16b}$$

where $\boldsymbol{D} = \text{Diag}(\boldsymbol{d})$ and the Kullback-Leibler (KL) divergence is employed as the distance measure. The reason is that $\boldsymbol{R}_{m,\ell}$ is a joint PMF of two random variables, and the KL-divergence is the most natural distance measure under such circumstances. Problem (16) is a nonconvex optimization problem, but can be handled by a simple alternating optimization procedure.

Specifically, we propose to solve the following subproblems cyclically:

$$\boldsymbol{A}_m \leftarrow \arg\min_{\mathbf{1}^\top\boldsymbol{A}_m=\mathbf{1}^\top,\ \boldsymbol{A}_m\geq\mathbf{0}} \sum_{\ell\in\mathcal{S}_m} \mathsf{KL}\left(\widehat{\boldsymbol{R}}_{m,\ell}||\boldsymbol{A}_m\boldsymbol{D}\boldsymbol{A}_\ell^\top\right) \tag{17a}$$

$$\boldsymbol{d} \leftarrow \arg\min_{\mathbf{1}^\top\boldsymbol{d}=1,\ \boldsymbol{d}\geq\mathbf{0}} \sum_{\ell\in\mathcal{S}_m} \mathsf{KL}\left(\widehat{\boldsymbol{R}}_{m,\ell}||\boldsymbol{A}_m\boldsymbol{D}\boldsymbol{A}_\ell^\top\right) \tag{17b}$$

where $\mathcal{S}_m$ denotes the index set of $\ell$'s such that $\boldsymbol{R}_{m,\ell}$ is available. Both of the above problems are convex optimization problems, and thus can be effectively solved via a number of off-the-shelf optimization algorithms, e.g., ADMM [18] and mirror descent [2]. The detailed summarized algorithm is in Algorithm 2. The alternating optimization algorithm is also guaranteed to converge to a stationary point under mild conditions [3, 32].

---

**Algorithm 2** `MultiSPA-KL`

---

   **Input:** Annotator Responses $\{X_m(\boldsymbol{f}_n)\}$.
   **Output:** $\widehat{\boldsymbol{A}}_m$ for $m = 1, \ldots, M, \widehat{\boldsymbol{d}}$.
   Estimate second order statistics $\widehat{\boldsymbol{R}}_{m,\ell}$;
   get initial estimates of $\{\widehat{\boldsymbol{A}}_m\}$ using `MultiSPA`
   **for** $t = 1$ **to** `MaxIter` **do**
     **for** $m = 1$ **to** $M$ **do**
       update $\boldsymbol{A}_m \leftarrow$ (17a);
     **end for**
     update $\boldsymbol{d} \leftarrow$ (17b);
   **end for**

---

Note that this coupled factorization formulation bears some resemblance to the coupled tensor factorization formulation in [37]. However, the two are very different in essence. The formulation in [37] relies on the third-order statistics to establish identifiability, while the formulation in (16) establishes identifiability using nonnegativity of the confusion matrices and the prior. The KL-divergence based fitting criterion also fits the statistical learning problem better than the least squares based criterion in [37].

## D Estimation of Prior Probability Vector

In this section, we discuss different methods to estimate the prior probability vector $\boldsymbol{d}$ once the confusion matrices are estimated via `MultiSPA` algorithm.

It is to be noted that the SPA-estimated $\widehat{\boldsymbol{A}}_m$ is up to column permutation, even if there is no noise, i.e., $\widehat{\boldsymbol{A}}_m = \boldsymbol{A}_m \boldsymbol{\Pi}_m$ in the best case. Since our algorithm runs SPA separately for different $\overline{\boldsymbol{Z}}_m$'s, the permutation matrices resulted by each run of SPA need not to be identical; i.e., it is highly likely that $\boldsymbol{\Pi}_\ell \neq \boldsymbol{\Pi}_m$ for $m \neq \ell$. To estimate the prior PMF $\boldsymbol{d}$, one will need to use estimators such as

$$\widehat{\boldsymbol{D}} = \widehat{\boldsymbol{A}}_m^{-1} \boldsymbol{R}_{m,\ell} (\widehat{\boldsymbol{A}}_\ell^\top)^{-1},$$

which cannot be applied before the permutation mismatch is fixed. In practice, the mismatch can be removed by a number of simple methods. For example, if annotator $\ell$ has co-labeled data with annotator $m$, then $\tilde{\boldsymbol{A}}_\ell = \boldsymbol{A}_\ell \boldsymbol{\Pi}_m$ can be estimated from $\boldsymbol{R}_{m,\ell}$ via $\tilde{\boldsymbol{A}}_\ell = \widehat{\boldsymbol{A}}_m^{-1} \boldsymbol{R}_{m,\ell}$. We also have $\widehat{\boldsymbol{A}}_\ell = \boldsymbol{A}_\ell \boldsymbol{\Pi}_\ell$ estimated from $\overline{\boldsymbol{Z}}_\ell$. Using a permutation matching algorithm, e.g., the Hungarian algorithm [20], one can easily remove the permutation mismatch between $\widehat{\boldsymbol{A}}_\ell$ and $\tilde{\boldsymbol{A}}_\ell$. Another more heuristic yet more efficient way is to rearrange the columns of $\widehat{\boldsymbol{A}}_m$ so that it is diagonally dominant—this makes a lot of sense if one believes that all the annotators are reasonably trained.

## E Geometry of The Sufficiently Scattered Condition

In this section, we present more discussion on the sufficiently scattered condition that is used in Theorem 3. To simplify the notation, we omit the superscript of $\boldsymbol{H}^{(i)}$ for $i = 1, 2$ and use $\boldsymbol{H}$ to denote these two matrices. The sufficiently scattered condition is geometrically intuitive. The key to understand this condition is the second-order cone $\mathcal{C}$, which is shown in Fig. 2. This cone is very special since it is tangent to all the facets of the nonnegative orthant.

A case where $\boldsymbol{H} \in \mathbb{R}^{N \times K}$ satisfies the sufficiently scattered condition is plotted in Fig. 3. One can see that the sufficiently scattered condition is much more relaxed compared to the condition that enables SPA (cf. Fig. 4). In order to apply SPA to $\overline{\boldsymbol{Z}}_m$, one needs that there are rows in $\overline{\boldsymbol{H}}_m$ that attain the extreme rays of the nonnegative orthant.

Figure 2: Illustration of the cone $\mathcal{C}$ in an 3-dimensional space.

Figure 3: A case where $\boldsymbol{H}$ satisfies the sufficiently scattered condition. The inner circle corresponds to $\mathcal{C}$, the dots correspond to $\boldsymbol{H}(q,:)$'s, and the triangle corresponds to the nonnegative orthant. The shaded region is $\mathsf{cone}\left\{\boldsymbol{H}^\top\right\}$.

## F    Proof of Theorem 1

### F.1    Identification Theory of SPA

To understand Theorem 1, let us start with a noisy nonnegative matrix factorization (NMF) model

$$\boldsymbol{X} = \boldsymbol{W}\boldsymbol{H}^\top + \boldsymbol{N}, \tag{18}$$

where $\boldsymbol{W} \in \mathbb{R}^{M \times K}$, $\boldsymbol{H} \in \mathbb{R}^{N \times K}$, $\boldsymbol{W} \geq \boldsymbol{0}$ and $\boldsymbol{H} \geq \boldsymbol{0}$, and $\boldsymbol{N}$ represents the noise. Also assume that $\mathrm{rank}(\boldsymbol{W}) = K$ and $\boldsymbol{H} = \boldsymbol{\Pi} \begin{bmatrix} \boldsymbol{I}_K \\ \boldsymbol{H}^* \end{bmatrix}$; i.e., there exists $\Lambda = \{q_1, \ldots, q_K\}$ such that $\boldsymbol{H}(\Lambda,:) = \boldsymbol{I}$. Also assume that $\boldsymbol{H}\boldsymbol{1} = \boldsymbol{1}$.

The SPA algorithm under this model is as follows [16, 10, 2, 4]:

$$\widehat{q}_1 = \arg\max_{q \in \{1,\ldots,N\}} \|\boldsymbol{X}(:,q)\|_2^2$$

$$\widehat{q}_k = \arg\max_{q \in \{1,\ldots,N\}} \left\|\boldsymbol{P}^\perp_{\widehat{\boldsymbol{W}}(:,1:k-1)}\boldsymbol{X}(:,q)\right\|_2^2, \ k > 1.$$

where

$$\widehat{\boldsymbol{W}}(:, 1:k-1) = [\boldsymbol{X}(:,\widehat{q}_1), \ldots, \boldsymbol{X}(:,\widehat{q}_{k-1})]$$

collects all the previously estimated columns of $\boldsymbol{W}$ and $\boldsymbol{P}^\perp_{\widehat{\boldsymbol{W}}(:,1:k-1)}$ is a projector onto the orthogonal complement of $\mathrm{range}(\widehat{\boldsymbol{W}}(:, 1:k-1))$.

Figure 4: A case where $\overline{\boldsymbol{H}}_m$ satisfies the condition for applying SPA (aka. the separability condition). The inner circle corresponds to $\mathcal{C}$, the dots correspond to $\overline{\boldsymbol{H}}_m(q,:)$'s, and the triangle corresponds to the nonnegative orthant. The shaded region is $\mathsf{cone}\left\{\overline{\boldsymbol{H}}_m^\top\right\}$.

When there is no noise, it was shown in the literature that SPA readily identifies $\Lambda$ [16, 2]. To see this, consider

$$\|\boldsymbol{X}(:,q)\|_2 = \left\|\sum_{k=1}^{K}\boldsymbol{W}(:,k)\boldsymbol{H}(q,k)\right\|_2 \le \sum_{k=1}^{K}\|\boldsymbol{W}(:,k)\boldsymbol{H}(q,k)\|_2$$

$$= \sum_{k=1}^{K}\boldsymbol{H}(q,k)\|\boldsymbol{W}(:,k)\|_2 \le \max_{k=1,\dots,K}\|\boldsymbol{W}(:,k)\|_2 ,$$

where the two equalities hold simultaneously if and only if $\boldsymbol{H}(q,:) = \boldsymbol{e}_k^\top$ for a certain $k$, i.e., $q \in \Lambda$. After identifying the first index $\widehat{q}_1$ in $\Lambda$, then by projecting all the data column onto the the orthogonal complement of $\boldsymbol{X}(:,\widehat{q}_1)$, the same $\boldsymbol{W}(:,k)$ will not come up again. Hence, $K$ steps of SPA identifies the whole $\boldsymbol{W}$.

A salient feature of SPA is that it is provably robust to noise. To be specific, Gillis and Vavasis have shown that:

**Lemma 1.** *[16] Under the described NMF model, assume that $\|\boldsymbol{N}(:,l)\|_2 \le \delta$ for all l. If the below holds:*

$$\delta \le \sigma_{\min}(\boldsymbol{W})\min\left(\frac{1}{2\sqrt{K-1}},\frac{1}{4}\right)\left(1+80\kappa^2(\boldsymbol{W})\right)^{-1},$$

*then, SPA identifies an index set $\widehat{\Lambda} = \{\widehat{q}_1,\dots\widehat{q}_K\}$ such that*

$$\max_{1\le j\le K}\min_{\widehat{q}_k\in\widehat{\Lambda}}\|\boldsymbol{W}(:,j)-\boldsymbol{X}(:,\widehat{q}_k)\|_2 \le \delta\left(1+80\kappa^2(\boldsymbol{W})\right)$$

*where $\kappa(\boldsymbol{W}) = \frac{\sigma_{\max}(\boldsymbol{W})}{\sigma_{\min}(\boldsymbol{W})}$ is the condition number of $\boldsymbol{W}$.*

### F.2  Proof of The Theorem

Since $\boldsymbol{R}_{m,\ell}$ is obtained by sample averaging of a finite number of pairwise co-occurrences, the estimated $\widehat{\boldsymbol{R}}_{m,\ell}$ is always noisy; i.e., we have

$$\widehat{\boldsymbol{R}}_{m,\ell} = \boldsymbol{R}_{m,\ell} + \boldsymbol{N}_{m,\ell}, \tag{20}$$

where the noise matrix $\boldsymbol{N}_{m,\ell}$ has same dimension as $\widehat{\boldsymbol{R}}_{m,\ell}$ or $\boldsymbol{R}_{m,\ell}$ and its norm can be bounded by Lemma 2.

**Lemma 2.** *[1] Let $\delta \in (0,1)$ and let $\widehat{\boldsymbol{R}}_{m,\ell}$ be the empirical average of S independent co-occurrences of random variables $X_m$ and $X_\ell$ where $X_m, X_\ell \in \{1,\dots,K\}$, then the following holds*

$$\mathsf{Pr}\left[\|\widehat{\boldsymbol{R}}_{m,\ell}-\boldsymbol{R}_{m,\ell}\|_F = \|\boldsymbol{N}_{m,\ell}\|_F \le \frac{1+\sqrt{\ln(1/\delta)}}{\sqrt{S}}\right] \ge 1-\delta$$

Using the estimates $\widehat{\boldsymbol{R}}_{m,\ell}$, $\widehat{\boldsymbol{Z}}_m$ is constructed according to (5) (with $\boldsymbol{R}_{m,\ell}$'s replaced by $\widehat{\boldsymbol{R}}_{m,\ell}$'s). The columns of $\widehat{\boldsymbol{Z}}_m$ are normalized before performing `MultiSPA`, essentially normalizing the columns of $\widehat{\boldsymbol{R}}_{m,\ell}$. Normalization complicates the analysis since the demonstrators used in this step are also noisy. We derive Lemma 3 to characterize the noise bound after column normalization.

**Lemma 3.** *Assume that there exists at least $S$ joint responses from each of the annotator pairs $m, \ell$. Let $\eta \in (0, 1)$. If $\|\widehat{\boldsymbol{R}}_{m,\ell}(:, k)\|_1 \geq \eta$ and $\|\boldsymbol{N}_{m,\ell}(:, k)\|_1 < \|\boldsymbol{R}_{m,\ell}(:, k)\|_1, \forall k \in \{1, \ldots, K\}, \forall m \neq \ell$, then with probability greater than $1 - \delta$, the below holds $\forall k \in \{1, \ldots, K\}, \forall m \neq \ell$,*

$$\frac{\widehat{\boldsymbol{R}}_{m,\ell}(:, k)}{\|\widehat{\boldsymbol{R}}_{m,\ell}(:, k)\|_1} = \frac{\boldsymbol{R}_{m,\ell}(:, k)}{\|\boldsymbol{R}_{m,\ell}(:, k)\|_1} + \overline{\boldsymbol{N}}_{m,\ell}(:, k)$$

*where $\|\overline{\boldsymbol{N}}_{m,\ell}(:, k)\|_2 \leq \frac{2\sqrt{K}(1+\sqrt{\ln(1/\delta)})}{\sqrt{S}\eta}$.*

*Proof.* For simpler representation, let us assign $\boldsymbol{x} := \boldsymbol{R}_{m,\ell}(:, k)$, $\widehat{\boldsymbol{x}} := \widehat{\boldsymbol{R}}_{m,\ell}(:, k)$ and $\boldsymbol{n} := \widehat{\boldsymbol{R}}_{m,\ell}(:, k) - \boldsymbol{R}_{m,\ell}(:, k) = \boldsymbol{N}_{m,\ell}(:, k)$.

Let $\boldsymbol{x} = [x_1, \ldots, x_K]^\top$ and $\boldsymbol{n} = [n_1, \ldots, n_K]^\top$. Note that $\boldsymbol{x} \geq 0$ and $\boldsymbol{x} + \boldsymbol{n} \geq 0$—since $\boldsymbol{x}$ is a legitimate PMF and $\boldsymbol{x} + \boldsymbol{n}$ is averaged from co-occurrence counts. Then, we have

$$\frac{\widehat{\boldsymbol{x}}}{\|\widehat{\boldsymbol{x}}\|_1} = \frac{\boldsymbol{x} + \boldsymbol{n}}{\|\boldsymbol{x} + \boldsymbol{n}\|_1} = \frac{\boldsymbol{x} + \boldsymbol{n}}{\sum_i x_i + n_i} = \frac{\boldsymbol{x} + \boldsymbol{n}}{\sum_i x_i + \sum_i n_i}$$
$$= \frac{\boldsymbol{x} + \boldsymbol{n}}{\sum_i x_i \left(1 + \frac{\sum_i n_i}{\sum_i x_i}\right)}.$$

Let $\mu = \frac{\sum_i n_i}{\sum_i x_i}$. Using the assumption $\|\boldsymbol{N}_{m,\ell}(:, k)\|_1 < \|\boldsymbol{R}_{m,\ell}(:, k)\|_1$, then $|\mu| < 1$, From this,

$$\frac{\boldsymbol{x} + \boldsymbol{n}}{\|\boldsymbol{x} + \boldsymbol{n}\|_1} = \frac{(\boldsymbol{x} + \boldsymbol{n})(1 + \mu)^{-1}}{\sum_i x_i} = \frac{(\boldsymbol{x} + \boldsymbol{n})}{\sum_i x_i}(1 - \mu + \mu^2 - \mu^3 + \ldots)$$
$$= \frac{\boldsymbol{x}}{\sum_i x_i} - \mu \frac{\boldsymbol{x}}{\sum_i x_i}(1 - \mu + \mu^2 - \mu^3 + \ldots) + \frac{\boldsymbol{n}}{\sum_i x_i}(1 - \mu + \mu^2 - \mu^3 + \ldots)$$
$$= \frac{\boldsymbol{x}}{\sum_i x_i} + \frac{\boldsymbol{n}}{(1 + \mu) \sum_i x_i} - \frac{\mu \boldsymbol{x}}{(1 + \mu) \sum_i x_i}$$
$$= \frac{\boldsymbol{x}}{\|\boldsymbol{x}\|_1} + \underbrace{\frac{\boldsymbol{n} - \mu \boldsymbol{x}}{(1 + \mu) \sum_i x_i}}_{\Gamma} \tag{21}$$

Now let us bound the term $\Gamma$:

$$\|\Gamma\|_1 := \left\| \frac{\boldsymbol{n} - \mu \boldsymbol{x}}{(1 + \mu) \sum_i x_i} \right\|_1 \leq \frac{\|\boldsymbol{n}\|_1}{\|\boldsymbol{x} + \boldsymbol{n}\|_1} + \frac{\|\sum_i n_i\|}{\|\boldsymbol{x} + \boldsymbol{n}\|_1}$$
$$\leq 2 \frac{\|\boldsymbol{n}\|_1}{\|\boldsymbol{x} + \boldsymbol{n}\|_1}$$
$$\leq 2 \frac{\|\boldsymbol{n}\|_1}{\eta}. \tag{22}$$

The first and second inequalities are due to Cauchy-Schwartz ineqality and the last inequality is by $\|\widehat{\boldsymbol{R}}_{m,\ell}(:, k)\|_1 \geq \eta$.

From Lemma 2, with probability greater than $1 - \delta$, the below holds,

$$\sum_{k=1}^K \|\boldsymbol{N}_{m,\ell}(:, k)\|_2^2 = \|\boldsymbol{N}_{m,\ell}\|_F^2 \leq \frac{(1 + \sqrt{\ln(1/\delta)})^2}{S}.$$

By norm equivalence, $\frac{\|\boldsymbol{N}_{m,\ell}(:, k)\|_1}{\sqrt{K}} \leq \|\boldsymbol{N}_{m,\ell}(:, k)\|_2$, Therefore,

$$\sum_{k=1}^K \|\boldsymbol{N}_{m,\ell}(:, k)\|_1^2 \leq \frac{K(1 + \sqrt{\ln(1/\delta)})^2}{S}$$
$$\implies \|\boldsymbol{N}_{m,\ell}(:, k)\|_1^2 \leq \frac{K(1 + \sqrt{\ln(1/\delta)})^2}{S}, \quad \forall k$$
$$\implies \|\boldsymbol{N}_{m,\ell}(:, k)\|_1 \leq \frac{\sqrt{K}(1 + \sqrt{\ln(1/\delta)})}{\sqrt{S}}, \quad \forall k \tag{23}$$

From (22) and (23),

$$\|\Gamma\|_1 = \|\overline{N}_{m,\ell}(:,k)\|_1 \leq \frac{2\sqrt{K}(1 + \sqrt{\ln(1/\delta)})}{\sqrt{S}\eta}$$

By norm equivalence, $\|\overline{N}_{m,\ell}(:,k)\|_2 \leq \|\overline{N}_{m,\ell}(:,k)\|_1$, then

$$\|\overline{N}_{m,\ell}(:,k)\|_2 \leq \frac{2\sqrt{K}(1 + \sqrt{\ln(1/\delta)})}{\sqrt{S}\eta}$$

This completes the proof of Lemma 3. $\qquad\qquad\qquad\qquad\qquad\qquad\qquad\qquad\square$

With the above lemmas, we are ready to characterize the accuracy of applying SPA to identify the Dawid-Skene model given the assumptions in Eq. (11).

Eq. (11) indicates that there exits a set of indices $\Lambda_q = \{q_1, \ldots, q_K\}$ such that

$$\overline{H}_m(\Lambda_q, :) = I_K + E \tag{24}$$

where $I_K$ is the identity matrix of size $K$ and $E$ is the error matrix with $\max_j |E(l,j)| = \|E(l,:)\|_\infty \leq \epsilon$. By norm equivalence, we have $\|E(l,:)\|_2 \leq \sqrt{K} E(l,:)\|_\infty \leq \sqrt{K}\epsilon$.

Without loss of generality, let us assume $\Lambda_q = \{1, \ldots, K\}$ and

$$\overline{H}_m = \begin{bmatrix} I_K + E \\ H_m^* \end{bmatrix}.$$

Now we have,

$$\overline{Z}_m = A_m \overline{H}_m^\top + \overline{N} \tag{25}$$
$$= A_m [I_K + E^\top \quad (H_m^*)^\top] + \overline{N} \tag{26}$$
$$= A_m [I_K \quad (H_m^*)^\top] + [A_m E^\top \quad 0] + \overline{N} \tag{27}$$

where $\overline{N} = [\overline{N}_{m,m_1}, \ldots, \overline{N}_{m,m_{T(m)}}]$ and the zero matrix $0$ has the same size as that of $H_m^*$

This model is similar to the noisy NMF model, i.e., $X = W H^\top + N$, where the noise matrix $N = [A_m E^\top \quad 0_m^*] + \overline{N}$. To be specifc, we have

$$N(:,l) = A_m E(l,:)^\top + \overline{N}(:,l) \tag{28}$$

Therefore, one can see that

$$\|N(:,l)\|_2 = \|A_m E(l,:)^\top + \overline{N}(:,l)\|_2 \tag{29}$$
$$\leq \|A_m\|_2 \|E(l,:)\|_2 + \|\overline{N}(:,l)\|_2 \tag{30}$$
$$\leq \sigma_{\max}(A_m)\sqrt{K}\epsilon + \frac{2\sqrt{K}(1 + \sqrt{\ln(1/\delta)})}{\sqrt{S}\eta} \tag{31}$$

where (30) is by the Cauchy-Schwartz inequality, (31) is by Lemma 3 and $\sigma_{\max}(A_m)$ is the largest singular value of matrix $A_m$.

Hence, we effectively have the same model as in (18). Applying Lemma 1, we see that if

$$\epsilon \leq \frac{1}{\sqrt{K}\kappa(A_m)} \min\left(\frac{1}{2\sqrt{K-1}}, \frac{1}{4}\right)(1 + 80\kappa^2(A_m))^{-1} - \frac{2(1 + \sqrt{\ln(1/\delta)})}{\sigma_{\max}(A_m)\sqrt{S}\eta}$$
$$= \mathcal{O}\left(\max\left(K^{-1}\kappa^{-3}(A_m), \sqrt{\ln(1/\delta)}(\sigma_{\max}(A_m)\sqrt{S}\eta)^{-1}\right)\right),$$

then, with probability at least $1 - \delta$, the SPA algorithm identifies the matrix $\boldsymbol{A}_m$ with an error bound given by

$$
\begin{aligned}
&\min_{\boldsymbol{\Pi}} \|\widehat{\boldsymbol{A}}_m \boldsymbol{\Pi} - \boldsymbol{A}_m\|_{2,\infty} \\
&= \max_{1 \leq j \leq K} \min_{\widehat{q}_k \in \widehat{A}_q} \left\| \boldsymbol{A}_m(:, j) - \overline{\boldsymbol{Z}}_m(:, \widehat{q}_k) \right\|_2 \\
&\leq \left( \sigma_{\max}(\boldsymbol{A}_m)\sqrt{K}\epsilon + \frac{2\sqrt{K}(1 + \sqrt{\ln(1/\delta)})}{\sqrt{S}\eta} \right) \left(1 + 80\kappa^2(\boldsymbol{A}_m)\right) \\
&= \mathcal{O}\left( \sqrt{K}\kappa^2(\boldsymbol{A}_m)\max\left( \sigma_{\max}(\boldsymbol{A}_m)\epsilon, \sqrt{\ln(1/\delta)}(\sqrt{S}\eta)^{-1} \right) \right).
\end{aligned}
$$

This completes the proof.

## G    Proof of Theorem 2

Assuming that the rows of $\overline{\boldsymbol{H}}_m$ are generated from the probability simplex uniformly at random, we now analyze under what conditions vectors close to all $K$ vertices of the probability simplex appear in the rows of $\overline{\boldsymbol{H}}_m$.

Let us denote the probability simplex as $\mathcal{X} = \{\boldsymbol{x} \in \mathbb{R}^K | \boldsymbol{x}^\top \boldsymbol{1} = 1, \boldsymbol{x} \geq 0\}$.

Let us consider an $\epsilon$-neighbourhood of the $k$-th vertex $\boldsymbol{e}_k$ denoted as $\mathcal{Q}_k(\epsilon)$ such that

$$
\mathcal{Q}_k(\epsilon) := \{\boldsymbol{q} \in \mathcal{X} | \|\boldsymbol{q} - \boldsymbol{e}_k\|_2 \leq \epsilon\}. \tag{32}
$$

Also denote a euclidean ball of radius $\epsilon$ centered at $\boldsymbol{e}_k$ as $\mathcal{B}(\boldsymbol{e}_k, \epsilon)$. Geometrically, the continuous set $\mathcal{Q}_k(\epsilon)$ can be considered as the intersection of the probability simplex $\mathcal{X}$ and the euclidean ball $\mathcal{B}(\boldsymbol{e}_k, \epsilon)$, i.e, $\mathcal{Q}_k(\epsilon) = \mathcal{X} \cap \mathcal{B}(\boldsymbol{e}_k, \epsilon)$ (see Fig. 5).

Figure 5: The big triangle represents the probability simplex $\mathcal{X}$ when $K = 3$, the dotted circles denotes the euclidean balls $\mathcal{B}(\boldsymbol{e}_k, \epsilon)$, the shaded region denotes $\mathcal{X} \cap \mathcal{B}(\boldsymbol{e}_k, \epsilon)$. The small triangles near the vertices has the same volume as the simplex having edge lengths $\epsilon$ denoted as $\mathcal{X}_\epsilon$

Suppose we are uniformly sampling a set $\mathcal{P}$ of size $s$ from the probability simplex $\mathcal{X}$ such that $\mathcal{P} := \{\boldsymbol{p}_1, \boldsymbol{p}_2, \ldots, \boldsymbol{p}_s\}$

Let us define an event $J_i$ such that for every $i \in \{1, \ldots, s\}$,

$$
J_i = \begin{cases} 1, & \text{if } \boldsymbol{p}_i \in \mathcal{Q}_k(\epsilon) \\ 0, & \text{otherwise} \end{cases} \tag{33}
$$

Consider the probability such that event $J_i$ happens,

$$\Pr(J_i = 1) = \frac{\text{vol}(\mathcal{Q}_k(\epsilon))}{\text{vol}(\mathcal{X})} \tag{34}$$

$$= \frac{\text{vol}(\mathcal{X} \cap \mathcal{B}(\boldsymbol{e}_k, \epsilon))}{\text{vol}(\mathcal{X})} \tag{35}$$

$$\geq \frac{\text{vol}(\mathcal{X}_\epsilon)}{\text{vol}(\mathcal{X})} \tag{36}$$

$$\geq \left(\frac{\epsilon}{\sqrt{2}}\right)^{K-1} \tag{37}$$

where $\mathcal{X}_\epsilon$ denotes the $(K-1)$-dimensional simplex which intersects the co-ordinate axes at $\frac{\epsilon}{\sqrt{2}}\boldsymbol{e}_k$, for every $k \in \{1, \ldots K\}$ and its volume is given by $\frac{(\epsilon/\sqrt{2})^{K-1}}{(K-1)!}$ [35]. (Note that the probability simplex $\mathcal{X}$ intersects the co-ordinate axes at $\boldsymbol{e}_k$ for every $k$). The inequality in Eq.(36) uses the geometric property that the volume of $\mathcal{X} \cap \mathcal{B}(\boldsymbol{e}_k, \epsilon)$ is greater than the volume of $\mathcal{X}_\epsilon$ (see Fig. 5).

Let us define the random variable $U = \sum_{i=1}^s J_i$ which denotes the number of samples in $\mathcal{P}$ which belongs to the set $\mathcal{Q}_k(\epsilon)$. Then,

$$\mathbb{E}[U] = \mathbb{E}[\sum_{i=1}^s J_i] = \sum_{i=1}^s \mathbb{E}[J_i] \tag{38}$$

$$= \sum_{i=1}^s \Pr(J_i = 1) = s\Pr(J_1 = 1) \tag{39}$$

$$\geq s \left(\frac{\epsilon}{\sqrt{2}}\right)^{K-1} \tag{40}$$

Now, if there exists at least one sample from set $\mathcal{P}$ which is in the $\epsilon$-neighbourhood of $k$-th vertex, ie the event $J_i$ happens at least once, then $U = \sum_{i=1}^s J_i \geq 1$. We are interested in finding the below probability,

$$\Pr(U \geq 1) = 1 - \Pr(U < 1) \tag{41}$$

$$= 1 - \Pr(U \leq 0) \tag{42}$$

$$= 1 - \Pr(U = 0) \tag{43}$$

So, our goal boils down to finding $\Pr(U \leq 0)$ and we will achieve this using Chernoff-Hoeffding bound.

**Lemma 4.** *[36] Let $J_1, \ldots, J_s$ be independent bounded random variables such that $J_i$ falls in the interval $[a_i, b_i]$ with probability one and let $U = \sum_{i=1}^s J_i$. Then for any $t > 0$,*

$$\Pr(U - \mathbb{E}[U] \leq -t) \leq e^{-2t^2/\sum_{i=1}^s (b_i - a_i)^2} \tag{44}$$

It follows that

$$\Pr(U \leq \mathbb{E}[U] - t) \leq e^{-2t^2/\sum_{i=1}^s (b_i - a_i)^2} \tag{45}$$

By assigning $\mathbb{E}[U] - t = 0$, we get $t = \mathbb{E}[U]$. Also, notice that in our case $b_i = 1$, $a_i = 0$, then

$$\Pr(U \leq 0) \leq e^{-\frac{2\mathbb{E}[U]^2}{s}} \tag{46}$$

$$\leq e^{-\frac{s\epsilon^{2(K-1)}}{2^{K-2}}} \tag{47}$$

Eq. (47) is obtained by using the inequality (40) and implies that, the probability such that the uniform sample $\mathcal{P}$ does not contain any points from $k$-th vertex is less than $e^{-\frac{s\epsilon^{2(K-1)}}{2^{K-2}}}$.

Now we have to find the corresponding probability that considers all the $K$ vertices.

For this, let us define events $E_k$ as follows,

$$E_k = \{\text{There exists no point } \boldsymbol{p} \text{ in the uniform sample set } \mathcal{P} \text{ such that } \boldsymbol{p} \in \mathcal{Q}_k(\epsilon)\} \qquad (48)$$

From Eq. (47), it is clear that $\Pr(E_k) \leq e^{-\frac{s\epsilon^{2(K-1)}}{2^{K-2}}}$. Since the points are uniformly sampled from the probability simplex $\mathcal{X}$, this bound is applicable for all $k \in \{1, \dots, K\}$.

Now let us define the event $E$ as below

$$E = \{\text{there exists at least one point in the uniform sample set } \mathcal{P} \text{ such that } \boldsymbol{p} \in \mathcal{Q}_k(\epsilon) \text{ for each } k\}$$

We can observe that $E = \bigcap_{k=1}^{K} \overline{E_k}$ where $\overline{E_k}$ is the complement of the event $E_k$.

Therefore,

$$\begin{aligned}
\Pr(E) = \Pr\left(\bigcap_{k=1}^{K} \overline{E_k}\right) &= \Pr\left(\overline{\cup_{k=1}^{K} E_k}\right) \\
&= 1 - \Pr\left(\bigcup_{k=1}^{K} E_k\right) \\
&\geq 1 - \sum_k \Pr(E_k) \\
&\geq 1 - K e^{-\frac{s\epsilon^{2(K-1)}}{2^{K-2}}} \qquad (49)
\end{aligned}$$

Eq. (49) implies that with probability greater than or equal to $1 - K e^{-\frac{s\epsilon^{2(K-1)}}{2^{K-2}}}$, the points from the $\epsilon$-neighbourhood of all the vertices are contained by set $\mathcal{P}$.

If $s$ represents the number of rows in $\overline{\boldsymbol{H}}_m$, then for $s \geq \frac{2^{K-2}}{\epsilon^{2(K-1)}} \log\left(\frac{K}{\rho}\right)$, with probability at least $1 - \rho$, a uniform sample from the probability simplex $\mathcal{X}$ will contain $\epsilon$-near-vertex points of all the $K$ vertices. Note that $s = (M-1)K$ where $M$ is the number of annotators. This provides a bound on the number of annotators needed. Specifically, if there exists at least $1 + \frac{2^{K-2}}{K\epsilon^{2(K-1)}} \log\left(\frac{K}{\rho}\right)$ annotators, then we have the conclusion of Theorem 2

## H  Proof of Theorem 3

To show this theorem, we will use the following Lemma:

**Lemma 5.** [17] *Consider a matrix factorization model $\boldsymbol{R} = \boldsymbol{P}_1 \boldsymbol{P}_2^\top$, where $\boldsymbol{R} \in \mathbb{R}^{M \times N}$, $\boldsymbol{P}_1 \in \mathbb{R}^{M \times K}$, $\boldsymbol{P}_2 \in \mathbb{R}^{N \times K}$, and $\text{rank}(\boldsymbol{P}_1) = \text{rank}(\boldsymbol{P}_2) = K$. If $\boldsymbol{P}_i \geq \boldsymbol{0}$ for $i = 1, 2$ and both $\boldsymbol{P}_1$ and $\boldsymbol{P}_2$ are sufficiently scattered, we have any $\widehat{\boldsymbol{P}}_1 \geq \boldsymbol{0}$ and $\widehat{\boldsymbol{P}}_2 \geq \boldsymbol{0}$ that satisfy $\boldsymbol{R} = \widehat{\boldsymbol{P}}_1 \widehat{\boldsymbol{P}}_2^\top$ must have the following form*

$$\widehat{\boldsymbol{P}}_1 = \boldsymbol{P}_1 \boldsymbol{\Pi} \boldsymbol{\Sigma}, \quad \widehat{\boldsymbol{P}}_2 = \boldsymbol{P}_2 \boldsymbol{\Pi} \boldsymbol{\Sigma}^{-1}, \qquad (50)$$

*where $\boldsymbol{\Pi}$ is a permutation matrix and $\boldsymbol{\Sigma}^{-1}$ is a diagonal nonnegative singular matrix.*

Lemma 5 addresses the identifiability of a conventional nonnegative matrix factorization (NMF) model. Simply speaking, if both latent factors of $\boldsymbol{R} = \boldsymbol{P}_1 \boldsymbol{P}_2^\top$ are sufficiently scattered, then the NMF of $\boldsymbol{R}$ is unique up to column permutation and scaling of the latent factors.

Now we start proving Theorem 3. Let us consider the following matrix $\boldsymbol{R}$:

$$\boldsymbol{R} = \begin{bmatrix} \boldsymbol{R}_{m_1,\ell_1} & \boldsymbol{R}_{m_1,\ell_2} & \cdots & \boldsymbol{R}_{m_1,\ell_{|\mathcal{P}_2|}} \\ \vdots & \vdots & \cdots & \vdots \\ \boldsymbol{R}_{m_{|\mathcal{P}_1|},\ell_1} & \boldsymbol{R}_{m_1,\ell_2} & \cdots & \boldsymbol{R}_{m_{|\mathcal{P}_2|},\ell_{|\mathcal{P}_2|}} \end{bmatrix}$$

It is readily seen that

$$\begin{aligned}
\boldsymbol{R} &= \begin{bmatrix} \boldsymbol{A}_{m_1} \\ \vdots \\ \boldsymbol{A}_{m_{|\mathcal{P}_1|}} \end{bmatrix} \boldsymbol{D}[\boldsymbol{A}_{\ell_1}^\top, \dots, \boldsymbol{A}_{\ell_{|\mathcal{P}_2|}}^\top] \\
&= \boldsymbol{H}^{(1)} \boldsymbol{D}(\boldsymbol{H}^{(2)})^\top.
\end{aligned}$$

It suffices to show that $\boldsymbol{R} = \boldsymbol{H}^{(1)}\boldsymbol{D}(\boldsymbol{H}^{(2)})^\top$ is unique up to column permutations of $\boldsymbol{H}^{(i)}$ for $i = 1, 2$. The reason is that if such uniqueness holds, then $\boldsymbol{A}_m$ for $m \notin \mathcal{P}_1 \cup \mathcal{P}_2$ can be identified with via solving

$$\boldsymbol{R}_{m,r} = \boldsymbol{A}_m \boldsymbol{D} \boldsymbol{A}_r^\top, \ r \in \mathcal{P}_1 \cup \mathcal{P}_2,$$

up to the same column permutation.

Note that since $\boldsymbol{D}$ and $\boldsymbol{A}_r$ have been identified from $\boldsymbol{R}$, solving $\boldsymbol{A}_m$ amounts to solving a system of linear equations, which has a unique solution under $\mathrm{rank}(\boldsymbol{D}) = \mathrm{rank}(\boldsymbol{A}_m) = K$. Under the assumption that all $m \notin \mathcal{P}_1 \cup \mathcal{P}_2$ are connected to a certain $r \in \mathcal{P}_1 \cup \mathcal{P}_2$ via $\boldsymbol{R}_{m,r}$, all the $\boldsymbol{A}_m$'s for $m \notin \mathcal{P}_1 \cup \mathcal{P}_2$ can be identified up to the same column permutation.

To show the identifiability of $\boldsymbol{R} = \boldsymbol{H}^{(1)}\boldsymbol{D}(\boldsymbol{H}^{(2)})^\top$, consider re-writing the right hand side as

$$\boldsymbol{R} = \underbrace{\boldsymbol{H}^{(1)}\boldsymbol{D}^{1/2}}_{\boldsymbol{P}_1}\underbrace{\boldsymbol{D}^{1/2}(\boldsymbol{H}^{(2)})^\top}_{\boldsymbol{P}_2^\top}$$

$$= \boldsymbol{P}_1\boldsymbol{P}_2^\top.$$

It suffices to show that $\boldsymbol{P}_1$ and $\boldsymbol{P}_2$ are unique up to column permutation and scaling, since the constraints on $\mathbf{1}^\top\boldsymbol{A}_m = \mathbf{1}^\top(\Rightarrow \mathbf{1}^\top\boldsymbol{H}^{(i)} = |\mathcal{P}_i|)$ removes the scaling ambiguity.

Assume that there is an alternative solution $\boldsymbol{R} = \widehat{\boldsymbol{P}}_1\widehat{\boldsymbol{P}}_2^\top$. By the fact $\mathrm{rank}(\boldsymbol{A}_m) = K$, we have

$$\mathrm{rank}(\boldsymbol{H}^{(i)}) = K \Rightarrow \mathrm{rank}(\boldsymbol{P}_i) = K.$$

Note that $\boldsymbol{H}^{(1)}$ and $\boldsymbol{H}^{(2)}$ are both sufficiently scattered. This directly implies that both $\boldsymbol{P}_1$ and $\boldsymbol{P}_2$ are sufficiently scattered, since column scaling of $\boldsymbol{H}^{(1)}$ and $\boldsymbol{H}^{(2)}$ does not affect the cone of their respective transposes, i.e.,

$$\mathrm{cone}\left\{(\boldsymbol{H}^{(i)})^\top\right\} = \mathrm{cone}\left\{(\boldsymbol{H}^{(i)}\boldsymbol{\Sigma})^\top\right\}$$

for any nonnegative, nonsingular and diagonal $\boldsymbol{\Sigma}$.

Then, by Lemma 5, it must hold that

$$\widehat{\boldsymbol{P}}_1 = \boldsymbol{H}^{(1)}\boldsymbol{D}^{1/2}\boldsymbol{\Pi}\boldsymbol{\Sigma}, \ \widehat{\boldsymbol{P}}_2 = \boldsymbol{H}^{(2)}\boldsymbol{D}^{1/2}\boldsymbol{\Pi}\boldsymbol{\Sigma}^{-1},$$

for a certain $\boldsymbol{\Pi}$ and $\boldsymbol{\Sigma}$. Nevertheless, $\boldsymbol{\Sigma}$ is automatically removed by the constraints $\mathbf{1}^\top\boldsymbol{A}_m = \mathbf{1}^\top$.

# I   Proof of Theorem 4

Let $\overline{\boldsymbol{H}}$ represents the matrix $\boldsymbol{H}$ with its rows normalized with respect to $\ell_1$-norm. Geometrically, $\overline{\boldsymbol{H}}$ can be viewed as the projection of the rows of $\boldsymbol{H}$ onto the $(K-1)$-probability simplex. (cf. Fig. 2).

Lin *et al.* provides a characterization to the spread of the rows of $\overline{\boldsymbol{H}}$ in the probability simplex using a measure called as uniform pixel purity level (named in the context of hyperspectral imaging) [27]. The purity level is denoted by $\gamma$ and is defined as follows:

$$\gamma = \sup\{r \leq 1 | \mathcal{R}(r) \subseteq \mathrm{conv}\{\overline{\boldsymbol{H}}^\top\}\} \tag{51}$$

where

$$\mathcal{R}(r) = \{\boldsymbol{x} \in \mathbb{R}^K | \|\boldsymbol{x}\|_2 \leq r\} \cap \mathrm{conv}\{\boldsymbol{e}_1, \ldots, \boldsymbol{e}_K\} \tag{52}$$

Geometrically, $\mathcal{R}(\gamma)$ is the 'largest' $\mathcal{R}(r)$ that can be inscribed inside $\mathrm{conv}\{\overline{\boldsymbol{H}}^\top\}$. There is an interesting link between $\gamma$ and the sufficiently scattered condition shown in [11]:

**Lemma 6.** *[11] Assume $K \geq 3$ holds. If $\gamma \geq \frac{1}{\sqrt{K-1}}$, then $\overline{\boldsymbol{H}}$ is sufficiently scattered.*

In general cases, it is hard to check if $\gamma \geq \frac{1}{\sqrt{K-1}}$ holds [17]. However, in [27], a sufficient condition for $\gamma \geq \frac{1}{\sqrt{K-1}}$ is derived:

**Lemma 7.** *[27] Suppose the following assumption holds true: for every $k, k^{'} \in \{1, \ldots, K\}, k \neq k^{'}$, there exist a row index $q_{kk'}$ in $\overline{H}$ such that*

$$\overline{H}(q_{kk'}, :) = \alpha_{kk'} e_k^{\top} + (1 - \alpha_{kk'}) e_{k'}^{\top} \tag{53}$$

*where $\frac{1}{2} < \alpha_{kk'} < 1$ for $K \geq 4$, $\frac{2}{3} < \alpha_{kk'} < 1$ for $K = 3$ and $\alpha_{kk'} = 1$, for $K = 2$. Then $\gamma \geq \frac{1}{\sqrt{K-1}}$ and by Lemma 6, $\overline{H}$ is sufficiently scattered.*

Figure 6: ($\varepsilon$-sufficiently scatterd) The big triangle represents the probability simplex $\mathcal{X}$ when $K = 3$, the shaded region denotes the region which is $\varepsilon$ near the edges and in the inset, the shaded region depicts the lower bound for the volume (area) of this region at each vertex.

Let us define $\alpha = \min\limits_{k,k' \in \{1,\ldots,K\}, k \neq k'} \alpha_{kk'}$. Lemma 7 states that for every edges of the probability simplex, if there exists at least one row in $\overline{H}$ which belongs to certain range in the edge which is of length $\alpha$ ($\alpha$-edge) (see Fig. 6), then the matrix $\overline{H}$ is sufficiently scattered.

Let us denote the probability simplex as $\mathcal{X} = \{x \in \mathbb{R}^K | x^{\top} \mathbf{1} = 1, x \geq 0\}$.

For each vertex, there exists $K - 1$ edges associated to it. Let us denote an $\varepsilon$-neighbourhood of $\alpha$-edge connecting the vertices $k$ and $k^{'}$ as $\widetilde{\mathcal{Q}}_{k,k'}(\varepsilon, \alpha)$. By the conditions in Lemma 7 and the definition of $\varepsilon$- sufficiently scattered (cf. Def. 2), it can be seen that, for every edges connecting $k$ and $k'$, if there exists at least one row in $\overline{H}$ belonging to $\widetilde{\mathcal{Q}}_{k,k'}(\varepsilon, \alpha)$, then $\overline{H}$ is $\varepsilon$-sufficiently scattered.

For each vertex $k$, the union of $\widetilde{\mathcal{Q}}_{k,k'}(\varepsilon, \alpha)$, $k' \neq k$ forms a continuous neighbourhood around the vertex $k$ denoted as $\widetilde{\mathcal{Q}}_k(\varepsilon, \alpha)$ , i.e,

$$\widetilde{\mathcal{Q}}_k(\varepsilon, \alpha) = \bigcup_{k' \neq k} \widetilde{\mathcal{Q}}_{k,k'}(\varepsilon, \alpha) \tag{54}$$

Geometrically, the volume of the continuous set $\widetilde{\mathcal{Q}}_k(\varepsilon, \alpha)$ can be lower bounded as below (see Fig 6)

$$\text{vol}(\widetilde{\mathcal{Q}}_k(\varepsilon, \alpha)) \geq \text{vol}(\mathcal{X}_\alpha) - \text{vol}(\mathcal{X}_{\alpha-2\varepsilon}) \tag{55}$$

where $\mathcal{X}_{\alpha'}$ is $(K - 1)$-dimensional simplex which intersects the co-ordinate axes at $\frac{\alpha'}{\sqrt{2}} e_k$ for every $k = \{1, \ldots, K\}$ and thus has the edge lengths $\alpha'$. The volume of $\mathcal{X}_{\alpha'}$ is given by $\frac{(\alpha'/\sqrt{2})^{K-1}}{(K-1)!}$ [35]

Eq. (55) can then be written as

$$\text{vol}(\widetilde{\mathcal{Q}}_k(\varepsilon, \alpha)) \geq \frac{\alpha^{K-1}}{\sqrt{2}^{K-1}(K-1)!} - \frac{(\alpha - 2\varepsilon)^{K-1}}{\sqrt{2}^{K-1}(K-1)!} \tag{56}$$

$$= \frac{\alpha^{K-1}}{\sqrt{2}^{K-1}(K-1)!}\left(1 - \left(1 - \frac{2\varepsilon}{\alpha}\right)^{K-1}\right) \tag{57}$$

$$= \frac{\alpha^{K-1}}{\sqrt{2}^{K-1}(K-1)!}\left(\frac{2\varepsilon}{\alpha}\left(1 + \left(1 - \frac{2\varepsilon}{\alpha}\right) + \cdots + \left(1 - \frac{2\varepsilon}{\alpha}\right)^{K-2}\right)\right) \tag{58}$$

$$= \frac{\alpha^{K-1}}{\sqrt{2}^{K-1}(K-1)!}\left(\frac{2\varepsilon}{\alpha} + \frac{2\varepsilon}{\alpha}\left(1 - \frac{2\varepsilon}{\alpha}\right) + \cdots + \frac{2\varepsilon}{\alpha}\left(1 - \frac{2\varepsilon}{\alpha}\right)^{K-2}\right) \tag{59}$$

$$\geq \frac{\alpha^{K-1}}{\sqrt{2}^{K-1}(K-1)!}\frac{2\varepsilon}{\alpha} \tag{60}$$

$$= \frac{\alpha^{K-2}\varepsilon}{\sqrt{2}^{K-3}(K-1)!} \tag{61}$$

Eq. (58) uses the geometric series sum formula $1 - a^n = (1 - a)(1 + a + \cdots + a^{n-1})$ and the assumption that $\alpha > 2\varepsilon$.

From Eq. (54) and (61), the volume of the set $\widetilde{\mathcal{Q}}_{k,k'}(\varepsilon, \alpha)$ can be lower bounded as

$$\text{vol}\left(\bigcup_{k' \neq k} \widetilde{\mathcal{Q}}_{k,k'}(\varepsilon, \alpha)\right) = \text{vol}\left(\widetilde{\mathcal{Q}}_k(\varepsilon, \alpha)\right) \tag{62}$$

$$\implies (K-1)\text{vol}\left(\widetilde{\mathcal{Q}}_{k,k'}(\varepsilon, \alpha)\right) \geq \text{vol}\left(\widetilde{\mathcal{Q}}_k(\varepsilon, \alpha)\right) \tag{63}$$

$$\geq \frac{\alpha^{K-2}\varepsilon}{\sqrt{2}^{K-3}(K-1)!} \tag{64}$$

$$\implies \text{vol}\left(\widetilde{\mathcal{Q}}_{k,k'}(\varepsilon, \alpha)\right) \geq \frac{\alpha^{K-2}\varepsilon}{\sqrt{2}^{K-3}(K-1)(K-1)!} \tag{65}$$

Suppose we are uniformly sampling a set $\mathcal{P}$ of size $s$ from the probability simplex $\mathcal{X}$ such that $\mathcal{P} := \{\boldsymbol{p}_1, \boldsymbol{p}_2, \ldots, \boldsymbol{p}_s\}$

Let us define an event $J_i$ such that for every $i \in \{1, \ldots, s\}$,

$$J_i = \begin{cases} 1, & \text{if } \boldsymbol{p}_i \in \widetilde{\mathcal{Q}}_{k,k'}(\varepsilon, \alpha) \\ 0, & \text{otherwise} \end{cases} \tag{66}$$

Consider the probability such that event $J_i$ happens,

$$\Pr(J_i = 1) = \frac{\text{vol}(\widetilde{\mathcal{Q}}_{k,k'}(\varepsilon, \alpha))}{\text{vol}(\mathcal{X})} \tag{67}$$

$$\geq (K-1)!\frac{\alpha^{K-2}\varepsilon}{\sqrt{2}^{K-3}(K-1)(K-1)!} \tag{68}$$

$$= \frac{\alpha^{K-2}\varepsilon}{(K-1)\sqrt{2}^{K-3}} \tag{69}$$

Eq. (68) uses the fact that the volume of the $(K-1)$-dimensional simplex $\mathcal{X}$ is given by $\frac{1}{(K-1)!}$ [35].

Now, let us define the random variable $U = \sum_{i=1}^{s} J_i$. Then,

$$\mathbb{E}[U] = \mathbb{E}[\sum_{i=1}^{s} J_i] = \sum_{i=1}^{s} \mathbb{E}[J_i] \tag{70}$$

$$= \sum_{i=1}^{s} \mathsf{Pr}(J_i = 1) = s\mathsf{Pr}(J_1 = 1) \tag{71}$$

$$\geq s \frac{\alpha^{K-2}\varepsilon}{(K-1)\sqrt{2}^{K-3}}. \tag{72}$$

Now, if there exists at least one sample from set $\mathcal{P}$ which is in the $\varepsilon$-neighbourhood of $\alpha$-edge, i.e, the event $J_i$ happens at least once, then $U = \sum_{i=1}^{s} J_i \geq 1$. Also,

$$\mathsf{Pr}(U \geq 1) = 1 - \mathsf{Pr}(U < 1)$$
$$= 1 - \mathsf{Pr}(U \leq 0)$$
$$= 1 - \mathsf{Pr}(U = 0)$$

So, our goal boils down to finding $\mathsf{Pr}(U \leq 0)$ and we will achieve this using Lemma 4.

From Lemma 4, it follows that

$$\mathsf{Pr}(U \leq \mathbb{E}[U] - t) \leq e^{-2t^2/\sum_{i=1}^{s}(b_i - a_i)^2} \tag{73}$$

By assigning $\mathbb{E}[U] - t = 0$, we get $t = \mathbb{E}[U]$. Also, notice that in our case $b_i = 1$, $a_i = 0$, then

$$\mathsf{Pr}(U \leq 0) \leq e^{-\frac{2\mathbb{E}[U]^2}{s}} \tag{74}$$

$$\leq e^{-\frac{s\alpha^{2(K-2)}\varepsilon^2}{2^{K-4}(K-1)^2}} \tag{75}$$

Eq. (75) is obtained by using the inequality $\mathbb{E}[U] \geq s\frac{\alpha^{K-2}\varepsilon}{(K-1)\sqrt{2}^{K-3}}$ as in Eq. (72) and implies that, the probability such that the uniform sample $\mathcal{P}$ does not contain any points from $\widetilde{\mathcal{Q}}_{k,k'}(\varepsilon, \alpha)$ is less than $e^{-\frac{s\alpha^{2(K-2)}\varepsilon^2}{2^{K-4}(K-1)^2}}$.

Now we have to find the corresponding probability that considers all the $(K-1)$ edges for each vertex $k$.

For this, let us define events $\widetilde{E}_{kk'}$ as follows,

$$\widetilde{E}_{kk'} = \{\text{There exists no point } \boldsymbol{p} \text{ in the uniform sample set } \mathcal{P} \text{ such that } \boldsymbol{p} \in \mathcal{Q}_{k,k'}(\varepsilon, \alpha)\} \tag{76}$$

From Eq. (75), it is clear that $\mathsf{Pr}(\widetilde{E}_{kk'}) \leq e^{-\frac{s\alpha^{2(K-2)}\varepsilon^2}{2^{K-4}(K-1)^2}}$. Since the points are uniformly sampled from the probability simplex $\mathcal{X}$, this bound is applicable for all $k, k' \in \{1, \ldots, K\}, k \neq k'$.

Now let us define the event $\widetilde{E}$ as below

$$\widetilde{E} = \{\text{there exists at least one point in the set } \mathcal{P} \text{ such that } \boldsymbol{p} \in \mathcal{Q}_{k,k'}(\varepsilon, \alpha) \text{ for all } k, k', k \neq k'\}$$

We can observe that $\widetilde{E} = \bigcap_{\substack{k,k' \\ k \neq k'}} \overline{\widetilde{E}_{kk'}}$ where $\overline{\widetilde{E}_{kk'}}$ is the complement of the event $\widetilde{E}_{kk'}$.

Therefore,

$$\mathsf{Pr}(E) = \mathsf{Pr}\left(\bigcap_{\substack{k,k' \\ k \neq k'}} \overline{\widetilde{E}_{kk'}}\right) = \mathsf{Pr}\left(\overline{\bigcup_{\substack{k,k' \\ k \neq k'}} \widetilde{E}_{kk'}}\right)$$

$$= 1 - \mathsf{Pr}\left(\bigcup_{\substack{k,k' \\ k \neq k'}} \widetilde{E}_{kk'}\right)$$

$$\geq 1 - \sum_{k,k'} \mathsf{Pr}(\widetilde{E}_{kk'})$$

$$\geq 1 - K(K-1)e^{-\frac{s\alpha^{2(K-2)}\varepsilon^2}{2^{K-4}(K-1)^2}} \tag{77}$$

Eq. (77) implies that with probability greater than or equal to $1 - K(K-1)e^{-\frac{s\alpha^{2(K-2)}\varepsilon^2}{2^{K-4}(K-1)^2}}$, the points from the $\varepsilon$-neighbourhood of all the $\alpha$-edges are contained by set $\mathcal{P}$. This essentially means that the rows of $\overline{\boldsymbol{H}}$ satisfy the assumption (53) with $\varepsilon$ accuracy and thus the $\varepsilon$-sufficiently scattered condition is achieved.

From Lemma 7, we get the lower bounds for $\alpha$ under various values of $K$:

$$\alpha > \alpha_{min}, \quad \alpha_{min} = \begin{cases} 1, & \text{for } K = 2, \\ \frac{2}{3}, & \text{for } K = 3, \\ \frac{1}{2}, & \text{for } K > 3. \end{cases} \tag{78}$$

Therefore, Eq. (77) can be agian bounded as,

$$\Pr(E) \geq 1 - K(K-1)e^{-\frac{s\alpha_{min}^{2(K-2)}\varepsilon^2}{2^{K-4}(K-1)^2}} \tag{79}$$

If $s$ represents the number of rows in $\overline{\boldsymbol{H}}$, then for $s \geq \frac{2^{K-4}(K-1)^2}{\alpha_{min}^{2(K-2)}\varepsilon^2}\log\left(\frac{K(K-1)}{\rho}\right)$ , with probability at least $1 - \rho$, $\boldsymbol{H}$ is $\varepsilon$-sufficienty scattered. Note that $s = (M-1)K$ where $M$ is the number of annotators. This provides a bound on the number of annotators needed.

Consequently, if there exists at least $1 + \frac{2^{K-4}(K-1)^2}{K\alpha_{min}^{2(K-2)}\varepsilon^2}\log\left(\frac{K(K-1)}{\rho}\right)$ annotators, then we have the conclusion of Theorem 4.

## J  Sample complexity for second order and third order statistics

In this section, we compare the sample complexity needed to estimate the second order statistics of the annotator responses from $m$ and $\ell$ denoted as $\boldsymbol{R}_{m,\ell}$ and the third order statistics of the annotator responses from $m$, $n$ and $\ell$ denoted as $\boldsymbol{R}_{m,n,\ell}$ given a dataset of $N$ samples to jointly label as one of the $K$ classes.

In crowdsourcing, not all samples are labeled by an annotator. To be specific, an annotator $m$ labels each sample with probability $p_m \in (0, 1]$ and in most of the practical cases, $p_m << 1$. For simpler analysis, let us take $p_m = p$, for all annotators. Then, this results to have an average of $\lceil Np^2 \rceil$ joint responses from annotators $m$ and $\ell$ and $\lceil Np^3 \rceil$ joint responses from annotators $m$, $n$ and $\ell$. With this and using the matrix and tensor concentration results from [39], the estimation error for $\boldsymbol{R}_{m,\ell}$ and $\boldsymbol{R}_{m,n,\ell}$ can be re-stated as, with probability at least $1 - \delta$,

$$\|\boldsymbol{R}_{m,\ell} - \widehat{\boldsymbol{R}}_{m,\ell}\|_F \leq \frac{1 + \sqrt{\log(1/\delta)}}{p\sqrt{N}} \tag{80}$$

$$\|\boldsymbol{R}_{m,n,\ell} - \widehat{\boldsymbol{R}}_{m,n,\ell}\|_F \leq \frac{1 + \sqrt{\log(K/\delta)}}{p^{\frac{3}{2}}\sqrt{N/K}} \tag{81}$$

It is clear from Eq. (80) and (81) that in order to achieve the same accuracy, third order statistics need much higher number of samples compared to second order statistics when $p$ is smaller and $K$ is larger.