[Reviews · NeurIPS 2019]

Reviewer 1



The authors propose a novel crowd-sourcing algorithmic framework based on the Dawid-Skene model. This includes two novel algorithms MultiSPA and MultiSPA-KL which both admit model identifiability under certain assumptions. Crucially the proposed framework, unlike recent work, depends only on second-order statistics thus leading to lower bounds on the amount of annotations required to achieve a certain error in approximating the confusion matrices. The authors have, in general, done a good job of positioning their work with respect to the state-of-the-art. However too much emphasis, especially given the paper's focus on complexity, is put on citation 37 which the proposed work is repeatedly compared too (e.g. lines 97,120). In fact I believe a more meaningful comparison is to citation 40, which though, as pointed out by the authors, also requires third-order statistics, uses group statistics. As pointed out by the authors of 40 this leads to an algorithm with an optimal convergence rate up to logarithmic factors. Perhaps the authors could comment on this in the rebuttal. As mentioned the paper proposed two novel algorithms which both enjoy the aforementioned theoretical properties under various assumptions. A positive aspect of the paper is that the second proposed algorithm is motivated by the aim to provide an alternative to the first in the case where the first's assumptions may not hold. Thus the two algorithms seem to complement each other in this respect. The paper involves a large body of theoretical analysis proving both model identifiability and improved convergence rates compared to prior art also based on Dawid-Skene. I should note here that I did not check the proofs of the theorems in detail. However assuming these are correct, I believe this to be the main strength of the paper. Regarding the empirical evaluation I would characterize it as sufficient. The proposed algorithms, and in particular MultiSPA-KL, are shown to outperform a number of (relevant) baselines on multiple datasets though the results are not impressive especially on the AMT datasets. For example MV -D&S seems to perform as well as MultiSPA and MultiSPA-D&S for a fraction of the run time. Spectral-D&S also seems competitive with the proposed methods ( Incidentally there seem to be some discrepancies between the results shown here for citation 40 and the results presented in table 2. of that paper). One point of concern here is that the algorithm were implemented in MATLAB which renders any arguments based on runtimes rather unconvincing; how fast something runs in MATLAB says little of how fast it would run if implemented with some view to efficiency.

Reviewer 2



## Crowdsourcing via Pairwise Co-occurences: Identifiability and Algorithms ### Summary The authors propose a new method to infer the parameters of the seminal model for label aggregation in the context of crowdsourcing proposed by Dawid-Skene. The novelty is that unlike the original inference based on the EM algorithm, the proposed inference algorithm, based on SPA, that comes with provable guarantees over the identifiability of the parameters, given that there is a good enough annotator in the pool of annotators. In addition, unlike recent methods that resort to high-order estimates to prove identifiability, the proposed algorithm only resort to pairwise statistics being more resilient to small and sparse annotation datasets that are ubiquitous in crowdsourcing. The authors derives finite sample bounds on the reconstruction error of the parameters of the annotator and sufficient conditions on the parameters of the annotators for these bounds to work. Overall, it is a strong paper with novel theoretical contributions and strong experimental results. The weakest point of the paper is that it fails to provide a better intuition about the behavior of the bound. ### Details The main idea of the paper is to reinterpret the model as a nonnegative matrix factorization problem and analyze the properties of the successive projection algorithm (SPA) that allows to solve this problem (under certain ideal conditions in a finite number of steps), in the context of the Dawid-Skene model. More specifically, in the presence of a perfect annotator (diagonal confusion matrix) the algorithm solves the problem in K steps, being K the number of classes. This is applied sequentially to estimate the confusion matrix of each annotator in what they call Multi-SPA. The most interesting part of the paper is the theoretical analysis summarized in four theorems whose goal is to see what happens when: a) The is no perfect annotator (an annotator with a diagonal matrix) b) The estimation of pairwise annotator probabilities (input to the algorithm) are based on a finite sample Theorem 1 proves a bound on the reconstruction error of the confusion matrix of each annotator given that, for every class, there exist an annotator whose probability mass is concentrated enough around the diagonal, and that the annotators have co-labelled a certain number of examples. In addition, in Theorem 2 they proof that the first condition is fulfill with high probability under certain conditions that justifies increasing the number of annotators. This is a nice quite result, however, I think the paper would greatly benefit from plotting this bounds and going deeper in the analysis of the dependency of the bound with respect to the different parameters. An analysis of these bounds for the particular datasets in the experimental section would be really interesting. Also, as a minor, the claim from lines 180-183 is confusion since the bound depends on the size of the co-labeled dataset, an therefore, the more sparse is the matrix, the worse would be the bound. I think this should be re-written for the purpose of clarity. In the second part of the paper, the authors propose a second algorithm to solve the problem base on a constrained linear program. The advantage is that they can relax the constraint of the existence of a perfect annotator in the MultiSPA algorithm, based on the definition of a sufficiently scatter confusion matrix (Theorem 3). Again, they show the dependency with the number of annotators, showing that we need a smaller number of annotators to guarantee the sufficiently scatter condition. However, the constraint linear problem is non convex, so it seems counterintuitive since the aforementioned theorems rely on finding the optimal solution of this problem. Nevertheless, they theoretical analysis is still valid and novel enough. Finally, the authors compare the proposed algorithms with a extensive set of baselines (methods and datasets) showing competitive performance. I think that even though the success in showing the good performance of the method, it seems a bit disconnected from the first part of the paper. Analyzing the behavior of the bounds in this practical scenarios would lead to nice insights that would improve the paper. Despite this, this is a well-written strong paper with nice theoretical and practical contributions.

Reviewer 3



Update: Thanks for the response. I've increased my score to 7. ------------------------------------------------------------------------------------- This work studies the well-known DS model for aggregating crowdsourced labels. Two SPA-based algorithms are proposed to estimate the parameters in the DS model. The first algorithm MultiSPA assumes that every worker has co-labelled some items with another worker whose confusion matrix is diagonal-dominant. The second algorithm MultiSPA-KL has theoretic guarantees on its performance under a more relaxed geometric assumption on the worker confusion matrices and the overall true label distribution. The authors conduct thorough experiments to compare the proposed algorithms with existing benchmarks on both synthetic and real-world datasets. The paper is well-organized. The identifiability problem is interesting and relevant. The proposed SPA-based algorithms are novel and have theoretic guarantees but only use second-order statistics unlike other tensor-based algorithms which requires three-order statistics. The experimental settings are described in full details. But the proposed algorithms don’t outperform existing methods significantly on real-world datasets. Some questions: The goal is inferring the true labels. Does the proposed MultiSPA algorithm have any theoretic guarantees on its performance, just like what Spectral-DS has in Zhang et al. (2014)? How well can the MultiSPA algorithm identify confusion matrices that are not diagonal-dominant? The name of minmax-entropy should be minimax-entropy, and it’s surprising that it achieves 91.61% error rate on TREC as shown Table 1 because TREC is a binary dataset. Could the authors check this or do the authors have any ideas why it fails?

[Author Response · NeurIPS 2019]

[**Reviewer 1- Baselines [37] and [40]**] Thank you for bringing this discussion up. We would like to clarify that, on a high level, the discussion regarding sample complexity (in lines 119-123) of the second-order statistics and third-order statistics applies to both the work in [40] (Zhang et al, 2014) and [37] (Traganitis et al, 2018). However, since [37] uses the third order statistics directly (without grouping the data) like what we do, it is more fair to compare with [37]. Since [40] needs to group the data first and then estimates certain "group third-order statistics", it may need more samples to obtain accurate estimates. We will add one remark on this subtle point in the final version.

[**Reviewer 1 - MATLAB-based Runtime**] We fully agree with the reviewer that MATLAB-based implementations may not exactly reflect the runtime performance in real systems. On the other hand, we hope that the runtime performance in the paper can serve as a useful reference—in case one would like to gain some insights (instead of the exact runtime) on the computational complexities of the algorithms. Nevertheless, we do agree with the reviewer on this point, and will add a remark to notify the readers.

| $\kappa(\boldsymbol{A}_m)$ | MSE | $K$ | MSE |
|---|---|---|---|
| 3.15 | 0.006 | 2 | 0.002 |
| 6.33 | 0.012 | 3 | 0.013 |
| 10.14 | 0.033 | 4 | 0.021 |
| 60.32 | 0.074 | 5 | 0.024 |
| 100.82 | 0.086 | 6 | 0.025 |

Figure 1: Synthetic-data experiments. MSE against $\kappa(\boldsymbol{A}_m)$ and $K$, respectively. $N = 10^4, p = 1, K = 3$; $\kappa(\boldsymbol{A}_m)$ is controlled by assigning $\boldsymbol{A}_m = \boldsymbol{I}_K + \beta * \mathtt{rand}(K, K)$, followed by column normalization and changing $\beta$; averaged over 10 random trials.

[**Reviewer 2 - More Insights on The Theorems**] Thank you for this nice suggestion. It is perhaps not easy to directly verify the theorems on real data since some of the problem parameters, such as $\varepsilon$, $\kappa(\boldsymbol{A}_m)$ and $\sigma_{\max}(\boldsymbol{A}_m)$, are hard to acquire. In our experiments, we change the parameter $p$ that directly affects the number of available samples $S$ for estimating the second order statistics; $p$ also affects $T(m)$, i.e., the number of annotators who co-label data with $m$. To gain more insights, we will also add a number of synthetic-data and real-data experiments in the supplementary materials; see, e.g., Figures 1-2. Again, thanks for this constructive comment.

[**Reviewer 2 - Lines 180-183**] The reviewer is correct: sparser annotator responses yield lower values of $S$ and thus the estimation error bound will get worse. We will re-write this part. In particular, "does not hurt" will be removed.

[**Reviewer 3 - Label Estimation Accuracy**] Thank you for this good point. To analyze the label estimation accuracy, one way is to adopt and modify the results in [40]. To be specific, after model identification, we employ a MAP predictor (see [37,40]) for label estimation. Let $y_n$ denote the true label of sample $n$. Assume that the conditions in Lemma 11 in [40] hold, and that $\boldsymbol{A}_m(k_m, k) \geq \nu$, for all $m, k_m, k$. In addition, assume that the MultiSPA-output estimates satisfy $\|\boldsymbol{A}_m - \widehat{\boldsymbol{A}}_m\|_\infty \leq \varphi = \min\left\{\frac{\nu}{2}, \frac{\nu\overline{D}}{16}\right\}$, for all $m$, where $\overline{D}$ is defined as in [40]. Then, if there exist at least $\widetilde{M} = \frac{4\log\, 2K}{\overline{D}}$ annotators, the MAP predictor yields $\widehat{y}_n = y_n$ for all $n$. Also notice that Theorem 2 in [37] can also be modified to characterize the label estimation accuracy using the models output by MultiSPA and MultiSPA-KL.

[**Reviewer 3 - Confusion Matrices without Diagonal Dominance**] Please note that MultiSPA and MultiSPA-KL do not need a particular $\boldsymbol{A}_m$ to be diagonal dominant. It only requires that, among annotators $m_1, \ldots, m_{T(m)}$, there exists at least one annotator who is specialized for class $k$ (i.e, who does not confuse class $k$ with other classes) for every $k = 1, \ldots, K$. Such annotators need not to have diagonal dominant confusion matrices; see Fig. 3. In our implementation, diagonal dominance was only used to fix the column permutation mismatches among the $\widehat{\boldsymbol{A}}_m$'s. But this is not the only way for fixing the mismatches. One can use the method as stated in Sec. D in the supplementary materials that does not need diagonal dominance. The method generally works; e.g., for MultiSPA on the Bluebird data, it outputs a classification error of 12.96% (while using diagonal dominance yields 13.88%); on the Web data, it gives 14.32% (15.22% using diagonal dominance). Nevertheless, we have observed that using diagonal dominance gives constantly good results over different datasets, while the method in Sec. D is not as stable (e.g., on the Dog data, 20.20% classification error v.s. 17.09% using diagonal dominance). Our understanding is that for real data, diagonal dominance is a reasonable assumption, and thus exploiting this structure may be beneficial. We will add these results.

[**Reviewer 3- Minimax-entropy Method**] We have observed that Minimax-entropy is also a strong candidate. However, the performance can be somewhat unstable especially when the annotator response data is very sparse. Our guess is that the objective function of the Minimax-entropy method involves some regularization parameters which are intended to prevent overfitting of the data as pointed out by the authors. For the TREC dataset that is very large but extremely sparse, the algorithm is somewhat sensitive to the regularization parameters—manually finding an 'optimal' regularization parameter is not easy and the results can be very far from being ideal from time to time.

Figure 2: Real-data experiment. Classification error against the number of samples $S$ on UCI 'Adult'.

Figure 3: An example where the confusion matrix is specialized for class 2, but not diagonally dominant; $\alpha, \varepsilon \in [0, 1]$.

[Meta-Review · NeurIPS 2019]

The reviewers are in agreement that this is an important contribution on a problem that comes up a lot in practice. the paper has good theoretical and practical results.